# THE HIDDEN COST OF WAITING FOR ACCURATE PREDICTIONS

**Ali Shirali**[*,1]**, Ariel Procaccia**[†,2]**, and Rediet Abebe**[†,3]

[1]University of California, Berkeley
[2]Harvard University
[3]ELLIS Institute, Max Planck Institute for Intelligent Systems, & Tübingen AI Center

## ABSTRACT

Algorithmic predictions are increasingly informing societal resource allocations by identifying individuals for targeting. Policymakers often build these systems with the assumption that by gathering more observations on individuals, they can improve predictive accuracy and, consequently, allocation efficiency. An overlooked yet consequential aspect of prediction-driven allocations is that of timing. The planner has to trade off relying on earlier and potentially noisier predictions to intervene before individuals experience undesirable outcomes, or they may wait to gather more observations to make more precise allocations. We examine this tension using a simple mathematical model, where the planner collects observations on individuals to improve predictions over time. We analyze both the *ranking* induced by these predictions and optimal *resource allocation*. We show that though individual prediction accuracy improves over time, counter-intuitively, the average ranking loss can worsen. As a result, the planner's ability to improve social welfare can decline. We identify inequality as a driving factor behind this phenomenon. Our findings provide a nuanced perspective and challenge the conventional wisdom that it is preferable to wait for more accurate predictions to ensure the most efficient allocations.

## 1 INTRODUCTION

Algorithmic predictions are playing a central role in societal resource allocation. Policymakers and organizations are increasingly turning to algorithmically-driven systems in contexts where resources are scarce in order to target resources with greater precision (Eubanks, 2018; Kleinberg et al., 2015; Kube et al., 2023; Mashiat et al., 2024; Perdomo et al., 2023; Toros & Flaming, 2018). Underpinning this growing reliance on predictions is the assumption that by gathering more observations about individuals over time, we can improve prediction accuracy and, consequently, allocation efficiency.

In practice, decisions around the *timing* of predictions and how they inform allocations reveal consequential trade-offs that the planner must navigate. On the one hand, the planner may wait to collect extensive data to refine their predictions before intervening. On the other hand, they can intervene early by relying on coarser data and noisier predictions. The potential advantage of the latter is that, in a fixed-horizon setting where the planner wants to prevent individuals from experiencing undesirable outcomes, the "window of opportunity" for this undesirable outcome to be realized closes. Furthermore, the underlying population changes with time, as those at greatest risk of experiencing such outcomes are more likely to "fail out" of the population early if they do not receive resources (Bierman, 2002; Abebe et al., 2020; Salganik et al., 2020). These factors pull in different directions, and it is not immediately apparent which factor dominates.

We examine this tension using a simple, versatile model where the planner predicts and intervenes on a population over time. Modeling a generic resource allocation problem, we assume the planner has a fixed budget of resources to prevent individuals from experiencing undesirable outcomes, such as eviction, job loss, poor health, or dropping out of school (Mashiat et al., 2024; Perdomo et al.,

---

[*]This work was conducted during a visit at Harvard University.
[†]Alphabetical order.

2023; Zezulka & Genin, 2024; Chan et al., 2012; Subbhuraam, 2021; Faria et al., 2017; Mac Iver et al., 2019). At each time step, the planner collects observations about individuals to improve their estimate of their underlying failure probability. The planner then uses the rankings induced by these estimates to allocate resources. Specifically, we ask:

1. *Ranking:* How does the ranking loss change as the planner collects more data, but some individuals fail out of the population?

2. *Allocation:* For a given instance of this problem, what is the optimal time to allocate resources? When is early intervention justified?

We present our results for two allocation problems: First, in a stylized setting, the planner is tasked with allocating all resources at once but can choose when to do so. We then use this as a building block to study the case where the planner can allocate resources over time. For both the ranking and allocation problems, we examine the role of inequality—as measured by the variance in the underlying failure probabilities—and surface it as a driving factor behind the optimal solutions.

We show that although individual prediction accuracy improves with more observations, counter-intuitively, this does not translate into improvements in the average ranking loss. To observe this, we decompose ranking loss into two counteracting effects: one due to improvements in prediction from additional observations and the second due to the change in population as individuals fail out of the active pool. We identify fundamental statistics that drive these two effects. We show that the change-in-population effect negatively impacts ranking performance and that this effect grows at least proportionally to the variance in the failure probabilities.

We then address both instantiations of the resource allocation problem. For the setting where the planner must allocate all resources simultaneously, we derive an upper bound on the optimal allocation time. We explicitly identify the roles of inequality and budget in expediting or deferring the optimal allocation time. We show that with high inequality or a large budget, allocating resources earlier results in greater social welfare. For the setting where the planner can allocate the budget over time, we design a provably optimal algorithm with a running time independent of the number of individuals. Using this algorithm, we then demonstrate that the optimal solution can concentrate the allocation around any time-point $t$, and it behaves consistently with our findings on one-time allocation.

Our results provide a nuanced perspective on the role of timing in prediction-driven allocations. In settings where the planner observes and intervenes on a population over time, they must balance the desire for more accurate predictions with the necessity for timely interventions. In the presence of significant inequality within the population, more accurate predictions do not necessarily lead to better ranking or improved allocations, providing a potential justification for early resource allocation.

For an extensive discussion on the related work and adjacent problem settings, refer to Appendix B.

## 2 MODEL AND PRELIMINARIES

In this section, we first introduce the notations necessary to present the basics of our model. We provide further notation, as needed, throughout the paper and summarize the key notations in Table 1.

We model the population over which the planner acts. We assume there is an initial population of $N$ individuals and consider a finite horizon setting where $t \in [1, T]$.[1] Each individual $i$ has some failure probability $p_i \in [0, 1]$, which captures their likelihood of failing out of the population between time steps. In the absence of intervention, this failure probability remains the same across time, and failure events of different individuals are independent.[2] Once an individual fails, they are no longer in the active pool of the population. We denote this active pool at time $t$ by $\mathcal{A}^t$.

**Prediction and ranking.** At each time step $t$, the planner observes a signal $o_i^t$ from each active individual $i \in \mathcal{A}^t$. These signals are analogous to observing loan or rental payments in housing and credit scoring, exam scores in education, and medical check-ups and tests in clinical settings. In our working model, these signals are drawn independently from a Bernoulli process

$$o_i^t \sim \text{Ber}(\tilde{p}_i) \,, \tag{1}$$

---

[1]Though we primarily consider the finite horizon setting, the key insights hold in the infinite horizon setting.
[2]If the problem has further structure, such as when students share a teacher, this assumption may not hold.

where $\tilde{p}_i$ is a function of $p_i$. We drop the explicit dependence on $p_i$ from $\tilde{p}(p_i)$ for ease of notation.

We assume that $\tilde{p}$ is an increasing function: The more likely an individual is to fail, the more likely we are to observe signals indicating this possibility. An individual $i$ will leave $\tilde{p}_i/p_i$ positive observations in expectation. Thus, a larger $\tilde{p}$ results in more observations from individuals before they fail out.

The planner is interested in the predictions as a means to rank and prioritize individuals. Given observations drawn from Eq. (1) and a prior over the failure probability, we will examine how the ranking risk of the Bayes' optimal ranking, measured on the active population, changes over time. This is the subject of Section 3.

**Targeting and allocation.** The planner has a budget $B$ of resources, such as housing vouchers, unemployment insurance, and preventive health screenings, to allocate to individuals in the active pool. We assume that assigning a resource to an individual has a fixed unit cost. We consider two common instantiations of allocation problems: We first study the *one-time allocation problem*, where the planner is tasked with finding the optimal time $t$ to allocate $B$. We then consider an *over-time allocation problem*, where the planner aims to find the optimal distribution of the budget across time.

To measure the efficiency of allocations, we define by $u^t(p)$ the expected utility the planner gets from intervening on an individual with failure probability $p$. The planner's primary objective is to maximize the utility over all treated individuals when there is no spillover effect. We assume that $u^t(p)$ is non-increasing in $t$ and non-decreasing in $p$, which captures the idea that the planner does not get more utility from intervening on the same individual later or from intervening on a better-off individual. We further assume that the utility function is concave in $p$, reflecting diminishing returns. Without loss of generality, we let $u^t(0) = 0$.

The utility framework we define above allows for significant generality. In particular, we note that for any individual $i$, the utility the planner gets from intervening on $i$ may depend on $i$'s unobserved characteristics, but this does not affect our results. Further, we make basic assumptions on $u^t$, allowing us to prove results in a general setting. To build intuition, at various points, we use a specific example of utilities, defined below.

*Example* 2.1 (Fully effective treatment). Suppose that once an individual receives a resource, they do not fail out of the population at subsequent time steps. The planner's utility then equals

$$u^t(p) = 1 - (1 - p)^{T-t}, \qquad (2)$$

i.e., allocating to individual has the same expected utility as the probability that this individual would have failed by time $T$ without this intervention.

For the allocation problems we consider, the planner uses the ranking induced by the predicted failure rates. We find that these rankings, in and of themselves, are interesting objects of study as they reveal complex trade-offs over time.

## 3 RANKING OVER TIME

In this section, we exclusively focus on ranking, which will form the basis for our main results on allocation presented in subsequent sections. We present our main ranking-related result separately both because it helps build intuition for the delicate tradeoff a planner needs to consider for the allocation problems, but also because it shows that these tradeoffs are present in other interventions that leverage risk-based rankings.

Informally, we show that even though individual predictions may improve as more observations become available over time, ranking quality can in fact decline. We demonstrate that inequality—as defined by the variance in the failure probabilities—characterizes such settings. The intuition is as follows: Although individual predictions improve over time, in high inequality settings, individuals with high $p$ (which were easier to distinguish from low $p$ individuals based on coarse information) are more likely to drop out earlier, leaving behind an active population that is harder to rank.

**Ranking risk.** We define ranking quality using ranking risk $R^t$ at time $t$. We consider a common notion of ranking risk based on pairwise ranking loss (Mohri, 2018). Given two individuals at time $t$, the pairwise ranking problem predicts which individual has the higher failure probability based on observations up to $t$. We assign a loss of zero if the prediction is correct and one otherwise.

We denote the number of positive observations from an active individual $i$ up to time $t$ by

$$y_i^t := \sum_{t' \in [t]} o_i^{t'} . \tag{3}$$

Then ranking individuals based on their $y^t$ is Bayes optimal in terms of the zero-one pairwise ranking loss.[3] Formally, the zero-one risk of optimal (pairwise) ranking at time $t$ is

$$R^t = \Pr_{i,j}^t \big( y_j^t < y_i^t \mid p_j \geq p_i \big) . \tag{4}$$

Here, $\Pr_{i,j}^t(\cdot)$ is the probability when we choose two active individuals from $\mathcal{A}^t$ independently.

**Main result.** We now present our main result on the dynamics of the optimal ranking risk. This result relies on certain approximations of the ranking risk, which we will detail later in this section.

**Theorem 3.1.** *The ranking risk of the optimal ranking at $t$ can only improve in the next time step if*

$$\underbrace{\frac{\mathrm{Var}^t[p]}{(1 - \mathbb{E}^t[p])^2} - O\big(\frac{1}{\sqrt{t}}\big)}_{\text{change-in-population effect}} < \underbrace{\frac{C_{\text{approx.}}}{t}}_{\text{gain in observations}} , \tag{5}$$

*where we assume that the inverse of $\tilde{p}(\cdot)$ is $O(1)$-Lipschitz. The proof presents the exact form of $O(1)$ and $C_{\text{approx.}}$ which we skip here for clarity of exposition.*

See proof on page 29. The necessary condition stated in Eq. (5) highlights two key insights into when ranking quality decreases over time. First, as $t$ increases, the left-hand side—which measures the change-in-population effect—increases, whereas the right-hand side—which measures the gain in observations—decreases. Second, as either the mean or variance in failure probability increases, it is again harder to satisfy Eq. (5). Combined, these insights highlight that later rankings are only preferred under conditions where inequality and average failure probabilities are low.

We next discuss the main steps towards proving Theorem 3.1.

**Approximating ranking risk.** To approximate ranking risk, we recall observations are drawn from $\mathrm{Ber}(\tilde{p})$. Eq. (3) then implies $y^t \sim \mathrm{Binomial}(t, \tilde{p})$. For analytic tractability, we approximate this with the normal distribution $\mathcal{N}$, with mean $t \cdot \tilde{p}$ and variance $t \cdot \tilde{\sigma}^2$. Here, $\tilde{\sigma}^2 := \tilde{p} \cdot (1 - \tilde{p})$. The independence of the draws in Eq. (4) implies

$$\frac{y_j^t - y_i^t}{t} \,\Big|\, (\tilde{p}_i, \tilde{p}_j) \,\sim\, \mathcal{N}\big(\tilde{p}_j - \tilde{p}_i, \frac{\tilde{\sigma}_{ij}^2}{t}\big) ,$$

where $\tilde{\sigma}_{ij}^2 := \tilde{\sigma}_i^2 + \tilde{\sigma}_j^2$. Denoting the cumulative distribution function of the standard normal distribution with $\Phi(\cdot)$, it is then straightforward to simplify $R^t$ as

$$R^t \approx \mathbb{E}_{i,j}^t \Big[ \Phi\big( - \frac{|\tilde{p}_j - \tilde{p}_i|}{\tilde{\sigma}_{ij}} \sqrt{t}\big)\Big] . \tag{6}$$

Note that the dependence of $R^t$ on $t$ appears in two places: inside $\Phi(\cdot)$, which captures the effect of gathering more observations over time, and $\mathbb{E}_{i,j}^t$, which models a change-in-population effect.

**Decomposing step-change in ranking risk.** We denote the change in $R^t$ in one time step by $\Delta R^t := R^{t+1} - R^t$. Using Eq. (6), we can decompose $\Delta R^t$ into two parts:

- The change in $\Phi$ after one step, which we approximate by taking the derivative with respect to $t$.
- The change of $\mathbb{E}_{i,j}^t$ after one step. To compute this, denote the distribution in failure probability of the population $\mathcal{A}^t$ by $\mathcal{P}^t$. Since an active individual at $t$ with a failure probability of $p$ survives until $t+1$ with a probability of $1-p$, we can write

  $$\mathcal{P}^{t+1}(p) = \big(\frac{1-p}{1-\mu^t}\big) \mathcal{P}^t(p) , \tag{7}$$

  where $\mu^t := \mathbb{E}^t[p]$. Using this, we can compute the expected change in population as follows:

  $$\mathbb{E}_{i,j}^{t+1}[\cdot] = \frac{\mathbb{E}_{i,j}^t\big[(1 - p_i)(1 - p_j)(\cdot)\big]}{(1 - \mu^t)^2} .$$

---

[3]Refer to Proposition E.4 for a proof.

Putting these two parts together, and approximating $\Phi(-x)$ with $\frac{1}{\sqrt{2\pi}x}\exp(-x^2/2)$, the change in ranking risk after one time step is

$$\Delta R^t = \mathbb{E}_{i,j}^t\left[\frac{1}{\sqrt{2\pi t}}\exp\left(\frac{-(\tilde{p}_j - \tilde{p}_i)^2}{2\tilde{\sigma}_{ij}^2}t\right)\left\{\frac{\tilde{\sigma}_{ij}}{|\tilde{p}_j - \tilde{p}_i|}\left[\frac{(1-p_i)(1-p_j)}{(1-\mu^t)^2} - 1\right] - \frac{|\tilde{p}_j - \tilde{p}_i|}{2\tilde{\sigma}_{ij}}\right\}\right]. \quad (8)$$

Intuitively, we can think of the exponential multiplier as a kernel enforcing similarity of $\tilde{p}_i$ and $\tilde{p}_j$. Such a filter pushes $\Delta R^t$ towards positive values, which means that as the filter becomes stricter for larger $t$, the effect of losing vulnerable individuals dominates the gain from collecting more observations on the remaining population. We make a mild assumption here that there exists a constant $\alpha \in (0, 1)$ bounded away from zero such that plugging $\exp(-x^2) \approx \mathbb{1}\{\exp(-x^2) \geq \alpha\}$ into Eq. (8) does not change the value of $\Delta R^t$. Then $C_{\text{approx.}} := \ln(1/\alpha)$. The rest of the proof of Theorem 3.1 is straightforward, which we leave in the appendix.

## 4 ONE-TIME ALLOCATION

In the previous section, we examined how ranking metrics evolve over time. In this section, we explore how these dynamics impact the allocation problem driven by the ranking.

The allocation policy varies depending on when resources become available and any spending restrictions in place. We assume a fixed budget for a specific population and explore two variations: In the general case, the budget can be allocated flexibly over time. For example, funding for a student cohort may be distributed across multiple years before graduation. In contrast, in the special case, the budget must be spent all at once, often due to administrative constraints, and the question becomes: when is the optimal time to allocate it? For example, cancer screenings may need to occur at a certain age, or school empowerment programs may be restricted to a specific grade due to the high cost of developing materials and training staff for multiple grades.

We first focus on the one-time allocation problem. We do so not only because it may represent a real-world constrained allocation problem but also because it provides valuable intuition for the more general variation of the problem. Importantly, strong theoretical results from this analysis offer key insights into the timing, budget, and inequality dynamics that form the core of our contribution. Informally, our main result states:

**Theorem** (Informal). *For the one-time allocation problem, there exists a $t^*$ that is a fraction of the horizon $T$, such that the planner can best optimize utility by allocating $B$ before $t^*$. This $t^*$ decreases, favoring earlier allocations, when inequality in the failure probabilities is high or the budget is large.*

To explicitly state this theorem, we need to introduce additional terminology, concepts, and definitions. Formally, given a budget $B$, at a chosen time $t$, the planner ranks active individuals $\mathcal{A}^t$ in descending order of their number of positive observations $y^t$, and allocates resources to the first $B$ of them.[4]

Denote the predicted ranking by injection $r^t : \mathcal{A}^t \to [N^t]$, where individual $i$ with $r^t(i) = 1$ is the most eligible one. The total utility of allocating a budget $B$ at time $t$ is given by

$$W^t = \sum_{i \in \mathcal{A}^t} u^t(p_i) \cdot \mathbb{1}\{r^t(i) \leq B\}.$$

Proving our results requires establishing an upper bound $t^*$ such that $W^t$ can increase over the next steps only if $t$ is less than $t^*$. In other words, deferring allocation is only justifiable if we have not reached this critical time. The results in this section show an intricate relationship between the timing of prediction/allocation with the inequality of the initial population and scarcity of resources.

Before presenting our results in full generality, we first consider the fully effective treatment setting, where the utility function follows Eq. (2). Although many real-world treatments may not be fully effective, many are, in fact, sufficiently effective for this special case to serve as a useful approximation for understanding their dynamics. For instance, consider housing vouchers or dropout prevention programs, which have been found to be very effective.[5] Due to its significance and analytical tractability, we next present specific results for this case.

---

[4]Proposition E.4 implies this is the Bayes optimal ranking in our setting.

[5]For instance, Gubits et al. (2016) found "significant positive impacts ... in families offered a voucher, and these impacts extended beyond housing stability."

## 4.1 THE FULLY EFFECTIVE TREATMENT SETTING

Recall from Example 2.1 that the utility function in the case of fully effective treatments follows $u^t(p) = 1 - (1-p)^{T-t}$. To state our results, we also need to define a measure of inequality:

**Definition 4.1** (G-decaying distribution). We say distribution $\mathcal{P}$ over $p$ is G-decaying for $G \geq 0$ if

$$- G \cdot \frac{\mathcal{P}(p)}{1-p} \leq \frac{\mathrm{d}\mathcal{P}}{\mathrm{d}p} \leq 0 \,.$$

Intuitively, $G$ bounds the rate at which the distribution decreases. Smaller values of $G$ correspond to *greater inequality* (see Proposition E.10 for the relationship between $G$ and $\mathrm{Var}[p]$). For example, if $\mathcal{P}(p) \propto (1-p)^{\beta-1}$—the probability density function of a $\mathrm{Beta}(1, \beta)$ distribution—then $\mathcal{P}$ is $(\beta - 1)$-decaying. Under this definition, the highest level of inequality in the population corresponds to the uniform distribution, which is 0-decaying.

Recall that at each time step $t$, the planner observes $o^t \sim \mathrm{Ber}(\tilde{p})$ from each active individual. To simplify notation, we assume the following concave observation model:[6]

*Assumption* 4.2 (Observation model). We assume observations from an individual with failure probability $p$ are independently drawn from $\mathrm{Ber}(\tilde{p})$, where $\tilde{p} = 1 - (1-p)^{\gamma}$, and $\gamma > 1$.

Under these assumptions, we present a necessary condition for when waiting for additional observations can improve the allocation efficiency of fully effective treatments:

**Theorem 4.3** (Conditions for fully effective treatment). *Suppose treatments are fully effective and the initial distribution over failure probabilities is G-decaying. The overall utility can improve after time $t$ only if $t < t^*$, where*

$$t^* := \frac{T}{2} + \left(\frac{G}{4} + \frac{\gamma + \gamma^{-1}}{4} + 1\right)\left((\gamma + 1)\ln(\frac{N}{B}) + 1\right) \,.$$

See proof on page 29.

This theorem implies that we rarely need to go beyond the halfway point of the time horizon to optimally allocate resources, and that this $t^*$ can be even smaller when we have high levels of inequality (corresponding to smaller $G$) or a larger budget. We further illustrate the intuition behind this theorem with an example in a simple setting in Appendix C. Combined, these results indicate that high levels of inequality and scarcity of resources play an important and consistent role in determining optimal allocation time—an insight we find carries over to more general settings.

## 4.2 THE GENERAL CLASS OF UTILITY SETTING

We define the class of $(\lambda_1, \lambda_2)$-decaying utility functions below. The fully effective treatment setting corresponds to a $(1, 1)$-decaying utility. Refer to Proposition E.11 for the proof and other examples.

**Definition 4.4** ($(\lambda_1, \lambda_2)$-decaying utility). We say a utility function $u^t(p)$ is $(\lambda_1, \lambda_2)$-decaying for positive $\lambda_1$ and $\lambda_2$ if at every $t$ and $p$,

$$u^{t+1}(p) \leq \frac{u^t(p) - \lambda_1 \cdot p}{1-p} \,, \qquad\qquad \text{(bounded decrease over time)}$$

$$(u^t)'(p) \leq \left(\frac{\lambda_2 - u^t(p)}{1-p}\right) \cdot (u^t)'(0) \,. \qquad\qquad \text{(bounded increase with } p)$$

When there is no finite $\lambda_2$ such that the second bound holds, we say that the utility is $\lambda_1$-decaying.

Our definition of $(\lambda_1, \lambda_2)$-decaying utility functions contains the essential elements to describe the utility function's behavior: a higher $\lambda_1$ indicates a stronger decrease over time, while a higher $\lambda_2$ signifies a faster increase with $p$. Our main result of this section states:

**Theorem 4.5** (Conditions for a general class of utilities). *For the general class of utilities, the overall utility can improve after time $t$ only if $t < t^* := (\gamma + 1)\ln(\frac{N}{B})$, or the following conditions hold:*

---

[6]We make the concavity assumption for ease of presentation, but as we discuss in the appendix, a weaker version of our result holds for any Lipschitz concave $\tilde{p}$ with bounded curvature.

1. *When the utility function is $(\lambda_1, \lambda_2)$-decaying with $\lambda_1 \geq \lambda_2$,*

$$(u^{t+1})'(0) \geq \left(\frac{\lambda_1}{\lambda_2}\right) \frac{t - t^*}{1 + (2 + \gamma + \gamma^{-1} + G)\frac{t^*+1}{2t-t^*+1}} \ .$$

2. *When the utility function is $(\lambda_1)$-decaying,*

$$(u^{t+1})'(0) \geq \lambda_1 \cdot \left(\frac{t - t^*}{t^* + 1}\right) \ .$$

See proof on page 30.

This theorem shows that for a wide-range of settings, the optimal allocation can happen early: Our assumptions about the concavity of the utility function and its zero value at $p = 0$ imply that $(u^{t+1})'(0)$ is a decreasing function of $t$. Conversely, regardless of whether the utility is $(\lambda_1, \lambda_2)$-decaying or just $\lambda_1$-decaying, the right-hand side of the bounds increases with $t$. Determining the timing of the optimal allocation is intertwined with the level of inequality and the budget. In particular, consistent with the previous section, a higher inequality and larger budget favor earlier allocations.

## 5 OVER-TIME ALLOCATION

The one-time allocation problem presented in the previous section helps as a building block for a more general setting, which we now consider. We study the problem of allocating a fixed budget over time to maximize total utility. First, we discuss the complexity of the problem under a naive optimization approach. Then, we characterize the optimal solution and present a method to find it regardless of the number of individuals. Through semi-synthetic experiments, we demonstrate that the optimal allocation follows a similar intuition to one-time allocation, where higher inequality or a larger budget shifts the optimal allocation toward earlier times.

### 5.1 CHARACTERIZING THE OPTIMAL OVER-TIME ALLOCATION

We begin by describing a Markov decision process (MDP) that governs the allocation problem. Let $\mathcal{A}_k^t$ denote the set of active individuals at time $t$ who have $k$ positive observations, i.e., $y^t = k$. We define the state at $t$ by $N_k^t := |\mathcal{A}_k^t|$, for $k \leq t$. An allocation policy specifies the individuals to treat from $\mathcal{A}_k^t$ at each state. The policy, together with the current state and the prior distribution over $p$, is sufficient to determine the distribution of the next state.

A naive approach to finding the optimal policy for the described MDP is as follows. Since individuals with a higher $y^t$ yield a higher expected utility,[7] the optimal allocation at every time $t$ should treat those with the highest $y^t$. Given a budget of $B$, we can specify a rollout of such a policy by the budget spent at each time step, resulting in $\binom{B+1}{T-1}$ possibilities. In the case of many agents, and thus a large $B$, the MDP dynamic becomes almost deterministic, and the optimal policy converges to a single fixed rollout. However, iterating over all $\binom{B+1}{T-1}$ rollouts to find the optimal policy is computationally infeasible. Next, we find a characterization of the optimal solution that significantly cuts down our search space.

**Theorem 5.1** (Optimal over-time allocation)**.** *The optimal over-time allocation in the limit of many individuals follows a specific pattern: For a non-decreasing sequence $q : [T] \rightarrow \{0, 1, \ldots, T\}$, there exists a time step $\hat{t} \in [T]$ such that,*

- *At $t \neq \hat{t}$, everyone with $y^t \geq q(t)$ will be treated. At the next step, $q(t + 1) \in \{q(t), q(t) + 1\}$.*

- *At $t = \hat{t}$, everyone with $y^t > q(t)$ and some with $y^t = q(t)$ will be treated. At the next step, $q(t + 1) \in \{q(t) + 1, q(t) + 2\}$.*

See proof on page 33.

This theorem narrows down the search space of possible policies to three parameters: $\hat{t}$, $q(\cdot)$, and what portion of $\mathcal{A}_{q(\hat{t})}^{\hat{t}}$ to treat. In particular, our search space no longer depends on the budget $B$ or the number of individuals.

---

[7]Refer to Lemma E.5 for a formal proof.

## 5.2 AN ALGORITHM TO FIND THE OPTIMAL SOLUTION

In this section, we present an algorithm that uses Theorem 5.1 to find the optimal policy, ensuring that its runtime does not scale with the number of individuals or the budget size. We build the algorithm to do so in pieces. First, we show that we can iterate over the joint space of unspecified parameters in $O(T^3 \cdot 2^{T-1})$ steps:

**Lemma 5.2.** *By specifying the time step $\hat{t} \in [T]$, the initial value of the sequence $q(\cdot)$ from the set of $\{0, 1, 2\}$, and a binary sequence of length $T - 1$, we can iterate over the joint space of unknown parameters in Theorem 5.1 in $O(T^3 \cdot 2^{T-1})$ steps.*

See proof on page 34.

This lemma utilizes the structure of sequence $q(\cdot)$ in Theorem 5.1 and reduces it to a binary sequence given $\hat{t}$. It also utilizes a linear structure of total utility to determine how many people to treat from $\mathcal{A}_{q(\hat{t})}^{\hat{t}}$. These are the first steps in Algorithm 1.

At the heart of Lemma 5.2 is Algorithm 2 that simulates a trajectory. This algorithm uses a backup formula that updates $N_k^t$ based on $N_k^{t-1}$ and $N_{k-1}^{t-1}$. Algorithm 2 also requires calculating expectation with respect to $p \sim \mathcal{P}^t(\cdot \mid y^t = k)$. In our simulation, this posterior has a closed form. However, in general, since $p$ is a bounded scalar, the posterior calculation can be well-approximated by a constant number of operations.

Given an efficient way to iterate over the search space, we next show that Algorithm 1 can find the optimal policy without increasing the run time:

**Theorem 5.3.** *Algorithm 1 finds the optimal over-time allocation in $O(T^3 \cdot 2^T)$ steps.*

See proof on page 34.

In real-world settings, time steps are typically on the scale of a month or a year. Therefore, $T$ is usually a small constant and the complexity of Algorithm 1 as stated in Theorem 5.3 is manageable. Compared to the naive iteration over $\binom{B+1}{T-1}$ possible trajectories, Algorithm 1's complexity is significantly reduced by dropping the dependency on the number of individuals and $B$. Powered by this algorithm, we next visualize the optimal over-time allocation in semi-synthetic settings.

---

**Algorithm 1** Optimal over-time allocation

1: $U_{\text{opt}} \leftarrow 0$
2: **for** $\hat{t} = 1$ to $T$, and $q(\cdot) \in$ valid sequences (as constructed in Lemma 5.2) **do**
    *Simulate as $\mathcal{A}_{q(\hat{t})}^{\hat{t}}$ are all treated:*
3:       $\{N_{q(t)}^t\}_{t=1}^T \leftarrow \text{SIMULATETRAJ}(1, \{N_0^1, N_1^1\}, q(\cdot))$
4:       $E_{\max} \leftarrow \mathbb{1}\{q(1) = 0\} \cdot N_1^1 + \sum_{t=1}^T N_{q(t)}^t$       ▷ maximum expenditure
    *Simulate the difference as if no one in $\mathcal{A}_{q(\hat{t})}^{\hat{t}}$ was treated:*
5:       $\{\Delta N_{q(t)}^t\}_{t=\hat{t}}^T \leftarrow \text{SIMULATETRAJ}(\hat{t}, \{0, \ldots, 0, N_{q(\hat{t})}^{\hat{t}}, 0, \ldots, 0\}, q(\cdot) + \mathbb{1}\{(\cdot) = \hat{t}\})$
6:       $\Delta E \leftarrow N_{q(\hat{t})}^{\hat{t}} - \sum_{t=\hat{t}+1}^T \Delta N_{q(t)}^t$       ▷ decrease from the max. expenditure
    *Find what proportion of $\mathcal{A}_{q(\hat{t})}^{\hat{t}}$ to treat:*
7:       $\rho \leftarrow \frac{E_{\max} - B}{\Delta E}$       ▷ proportion of $\mathcal{A}_{q(\hat{t})}^{\hat{t}}$ to be left untreated
8:     **if** $\rho > 1$ or $\rho < 0$ **then** continue to the next possible $q(\cdot)$ **end if**

    *Calculate the total utility:*
9:       **for** $t = 1$ to $T$ and $k = q(t)$ **do** $U_k^t \leftarrow \mathbb{E}_{p \sim \mathcal{P}^t(\cdot \mid y^t = k)}[u^t(p)]$ **end for**
10:      $U \leftarrow (1 - \rho) N_{q(\hat{t})}^{\hat{t}} U_{q(\hat{t})}^{\hat{t}} + \sum_{t \neq \hat{t}} (N_{q(t)}^t + \rho \Delta N_{q(t)}^t) U_{q(t)}^t + \mathbb{1}\{q(1) = 0\} \cdot N_1^1 U_1^1$

11:     **if** $U > U_{\text{opt}}$ **then** $U_{\text{opt}} \leftarrow U$, $q_{\text{opt}} \leftarrow q$, $\hat{t}_{\text{opt}} \leftarrow \hat{t}$ **end if**       ▷ check for optimality
12: **end for**
13: **return** $U_{\text{opt}}, q_{\text{opt}}, \hat{t}_{\text{opt}}$

---

## 5.3 VISUALIZING THE OPTIMAL SOLUTION

Algorithm 1 allows us to efficiently find the optimal over-time allocation. Using this algorithm, we next examine the effect of inequality, as encoded in the prior distribution of $p$, and the budget size on the optimal over-time allocation.

Our goal is to demonstrate that even in the simplest non-contrived settings, the tradeoff between gaining more observations and losing vulnerable individuals strongly exists. Therefore, we make minimal assumptions about the model: Suppose the treatments are fully effective, the observation model is $\tilde{p} = p$, and the prior over failure probabilities follows $\mathrm{Beta}(\alpha, \beta)$. The beta distribution is a common and expressive choice for modeling priors over $[0, 1]$-bounded random variables. It is also the conjugate prior for the binomial distribution, which allows us to write the posterior distribution over failure probability in closed form: $\mathcal{P}^t(\cdot \mid y^t = k) = \mathrm{Beta}(\alpha + k, \beta + 2t - k)$.

**Data and parameters estimation.** To make our simulations more realistic, we choose the initial distribution to reflect real-world data. In particular, we use National Education Longitudinal Study (NELS) of 1988, a longitudinal study with follow-ups at four points throughout the students' education.[8] In this data, failure corresponds to student dropout and is recorded.

We estimate the beta distribution parameters in the following manner. Let $m_0$ and $m_1$ be the proportion of the initial pool of individuals who failed right before the first and second steps, respectively. Assuming there has been no intervention at the first step, it follows from the central limit theorem that $m_0 \to \frac{\alpha}{\alpha+\beta}$ and $m_1 \to \frac{\alpha(\alpha+1)}{(\alpha+\beta)(\alpha+\beta+1)}$ at a fast rate of $O(1/\sqrt{N})$. By solving for $\alpha$ and $\beta$, we can accurately estimate the initial distribution as a Beta distribution. Our estimation gives $\alpha = 0.028$ and $\beta = 0.35$ for the case of NELS data. The mean of the estimated distribution also aligns with the NELS-provided estimation of dropout probability, with both methods predicting a dropout rate of around $7\%$.

**Results.** We first study the effect of budget size. Theorem 4.3 suggests that in case of one-time allocation, the optimal allocation time shifts with $\ln(\frac{N}{B})$. Fig. 1 suggests that a similar trend holds true in the case of over-time allocation: a larger budget favors earlier allocations.[9]

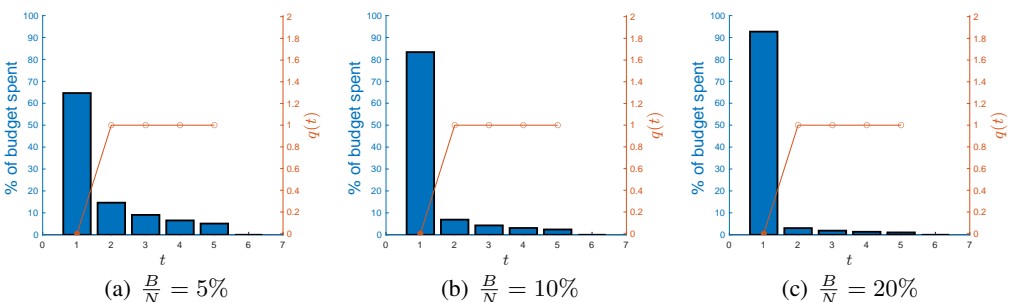

Figure 1: Optimal over-time allocation for three sizes of the budget and a fixed prior estimated from NELS data. The orange curve depicts the optimal $q(\cdot)$ and the filled circle corresponds to $t = \hat{t}$.

We next extend our analysis beyond the NELS data distribution to study the effect of initial distribution, particularly inequality, on the optimal allocation. Theorem 4.3 suggests that greater inequality can further favor earlier allocation. To simulate this effect, we fix $\frac{B}{N}$ at $10\%$ and consider three priors with differing tail decay. Fig. 2 indicates that as the prior approaches a uniform distribution, corresponding to maximum inequality in terms of our definition of $G$-decaying distributions, optimal allocation significantly favors earlier times.

These visualizations confirm that a similar intuition as the one-time allocation setting also appears in the over-time allocation problem.

---

[8] https://nces.ed.gov/surveys/nels88/

[9] Code is available at
https://github.com/alishiraliGit/hidden-costs-of-waiting-for-accurate-predictions.

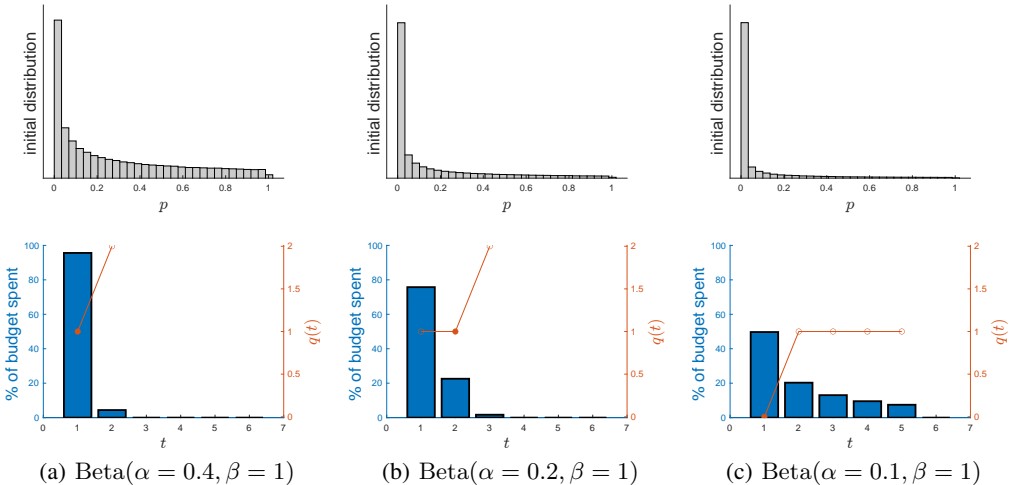

(a) Beta($\alpha = 0.4, \beta = 1$)     (b) Beta($\alpha = 0.2, \beta = 1$)     (c) Beta($\alpha = 0.1, \beta = 1$)

Figure 2: Optimal over-time allocation for three different priors and a fixed $\frac{B}{N} = 10\%$. The orange curve depicts the optimal $q(\cdot)$ and the filled circle corresponds to $t = \hat{t}$.

## 6    DISCUSSION

Our work contributes a timing dimension to an emerging body of research on evaluating prediction-driven allocation. Predictive systems are introduced with the promise of minimizing waste and increasing efficiency. This existing predisposition, amplified further by the traditional focus on maximizing predictive accuracy, encourages practices that favor waiting to collect more information over acting early with noisier signals. Our study presents a simple model that challenges this practice.

Our work opens numerous lines of inquiry. For instance, we assume the planner has a fixed budget $B$, corresponding to a fixed unit-cost intervention, that they can allocate all at once or over time. There are various natural variations worth exploring. For instance, we could consider heterogeneity in cost across time or different $p_i$ values. In the same spirit as Perdomo (2024), we can also consider trading off this $B$ with other interventions or parameters in the problem. We also think the tradeoffs between acting early and waiting to reduce uncertainty extend beyond our welfare-maximizing framework. For instance, future work could adapt our dynamic model to fairness-focused frameworks for studying uncertainty, such as those developed by Singh et al. (2021), and explore whether information gains consistently lead to improved outcomes.

We make generic assumptions about the failure probabilities and collection of observations. In settings motivating our study, the failure probabilities change over time, favoring increasing inequality in the absence of interventions. Likewise collecting observations for vulnerable individuals may be more costly, contain less signal, or may otherwise be undesirable (Monteiro Paes et al., 2022). Another general assumption we made is that, while individuals have heterogeneous values of $p$, we do not account for variations in their initial conditions or "starting points." Enriching the model we study to include such insights would only further justify early interventions in the presence of high inequality, though it would be interesting to examine the extent to which it does so. We should also point out that we consider the allocation problem in an unconstrained setting to highlight the generality of the tradeoffs. However, practical constraints can further limit the optimal allocation, in one way or another.

Our work introduces a potential lens through which to examine tradeoffs incurred by waiting to improve prediction accuracy. Our results, on their own, do not endorse early or late allocations for any specific setting. Each policy problem should be examined empirically, and policymakers must consider various community, policy, and practical considerations. Indeed, targeting as an effective means of improving welfare—which has fueled the use of predictive systems—is, itself, an actively debated policy concept (Shirali et al., 2024a). Nonetheless, we believe that the machine learning community can contribute to discussions around how to best evaluate predictive systems in such policy settings.

ACKNOWLEDGMENTS

We thank Jackie Baek, Joshua Blumenstock, Kate Donahue, Avi Feller, Moritz Hardt, Michael I. Jordan, and Jiduan Wu for their generous feedback and discussions. Abebe was partially supported by the Andrew Carnegie Fellowship. Procaccia was partially supported by the National Science Foundation under grants IIS-2147187 and IIS-2229881; by the Office of Naval Research under grants N00014-24-1-2704 and N00014-25-1-2153; and by a grant from the Cooperative AI Foundation.

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

# A    NOTATIONAL CONVENTIONS

The variables related to an individual $i$ are indexed with a subscript $i$. The variables related to time $t$ are indexed with a superscript $t$. The variables transformed with $\tilde{p}(\cdot)$ are denoted with a tilde.

| Symbol | Notion |
| --- | --- |
| $t$ | Time step which takes a value from $1$ to $T$ |
| $T$ | Horizon |
| $p_i$ | Failure probability of individual $i$ |
| $o_i^t$ | Binary observation from an active individual at time $t$ |
| $\epsilon$ | Noise on observation model of Eq. (1) |
| $\tilde{p}$ | $\tilde{p}(p) := (1 - \epsilon)f(p) + (1 - f(p))\epsilon$ |
| $y_i^t$ | Number of positive observations from an active individual $i$ up to time $t$: $y_i^t := \sum_{t' \in [t]} o_i^{t'}$ |
| $\mathcal{A}^t$ | Set of active individuals at time $t$ |
| $\mathcal{A}_k^t$ | Set of active individuals at $t$ with $y^t = k$: $\mathcal{A}_k^t := \{i \mid y_i^t = k\}$ |
| $\mathcal{I}^t$ | Set of individuals treated at $t$ |
| $\bar{\mathcal{A}}_k^t$ | $\mathcal{A}_k^t$ excluding those who will be treated at $t$: $\bar{\mathcal{A}}_k^t := \mathcal{A}_k^t \setminus \mathcal{I}^t$ |
| $N$ | Number of initial individuals at time $t = 1$ |
| $N^t$ | Number of individuals who made it to time $t$: $N^t := |\mathcal{A}^t|$ |
| $n^t$ | Proportion of initial individuals who made it to time $t$: $n^t := N^t/N$ |
| $n_k^t$ | $n_k^t := |\mathcal{A}_k^t|/N$ |
| $\bar{n}_k^t$ | $\bar{n}_k^t := |\bar{\mathcal{A}}_k^t|/N$ |
| $\mathcal{P}^t(\cdot)$ | Posterior over $p$ for an active individual at $t$ |
| $\mathcal{P}_k^t = \mathcal{P}^t(\cdot \mid y^t = k)$ | Posterior over $p$ given an individual has made it to $t$ and $y^t = k$ |
| $\mathrm{Pr}^t(\cdot)$ | Probability measure over active individuals at time $t$ |
| $\mathrm{Pr}_{i,j}^t(\cdot)$ | Probability measure over two independent active individuals $i$ and $j$ at time $t$ |
| $\mathrm{Pr}^t(y^t = k)$ | Probability that an active individual at $t$ has $y^t = k$ |
| $\mathrm{Pr}^t(y^t = k \mid p)$ | Likelihood that an active individual with failure probability $p$ shows $y^t = k$ |
| $\mathbb{E}^t[\cdot]$ | Expectation over active individuals at time $t$ |
| $\mathbb{E}_k^t[\cdot]$ | Expectation over active individuals at time $t$ with $y^t = k$ |
| $\mathbb{E}_{i,j}^t[\cdot]$ | Expectation over two independent active individuals $i$ and $j$ at time $t$ |
| $\mu^t$ | Mean of failure probability at time $t$: $\mu^t := \mathbb{E}^t[p]$ |
| $\mu_k^t$ | Mean of failure probability at time $t$ given $y^t = k$: $\mu^t := \mathbb{E}_k^t[p]$ |
| $\mathrm{Var}^t[\cdot]$ | Variance under $\mathrm{Pr}^t(\cdot)$ |
| $\tilde{\sigma}_i^2$ | $\tilde{\sigma}_i^2 := \tilde{p}_i \cdot (1 - \tilde{p}_i)$ |
| $\tilde{\sigma}_{ij}^2$ | $\tilde{\sigma}_{ij}^2 := \tilde{\sigma}_i^2 + \tilde{\sigma}_j^2$ |
| $G(\cdot)$ | Cumulative distribution function (CDF) of the standard normal distribution |
| $g(\cdot)$ | Probability density function (PDF) of the standard normal distribution |

Table 1: Glossary

## B  RELATED WORK

**Prediction for allocation.**  Algorithmic predictions are increasingly employed to identify individuals who are most in need of limited resources. In many of these applications, the timing of prediction and allocation is of the utmost importance. Examples include directing assistance to tenants at risk of eviction based on their predicted risk (Mashiat et al., 2024), using early warning systems to identify students at risk of dropout (Faria et al., 2017; Mac Iver et al., 2019; Perdomo et al., 2023; Rismanchian & Doroudi, 2023), prioritizing homelessness assistance while considering population dynamics (Toros & Flaming, 2018; Azizi et al., 2018; Kube et al., 2023), making ICU discharge decisions based on readmission or mortality probability (Chan et al., 2012; Shirali et al., 2024b), and improved targeting of humanitarian aids (Aiken et al., 2022). For a discussion on the role of machine learning in clinical medicine in particular, refer to Obermeyer & Emanuel (2016).

The adoption of predictive tools in resource allocation often comes with a promise that improvements in prediction accuracy can transfer to the allocation setting. Recent critical studies, however, have challenged this (Barabas et al., 2018; Shirali et al., 2024a; Perdomo, 2024). Our work gives a new timing dimension to this problem where prediction improvement is entangled with the population dynamics. Our work also emphasizes the role of inequality in this dynamic setting. In line with Shirali et al. (2024a) we found inequality a determinant factor in deciding which form of allocation works best in the interest of social welfare.

Welfare-maximizing treatment assignment under budget constraints is also a well-established topic in economics (Bhattacharya & Dupas, 2012; Kitagawa & Tetenov, 2018). While much of the literature focuses on estimating static heterogeneous treatment effects for a fixed population, we extend this work by examining the problem's dynamic aspects. Our model also bypasses the need to estimate treatment effects in observational settings (Athey & Wager, 2021) by incorporating these complexities into a general class of utility function and emphasizing the often-overlooked role of timing in predictions and allocations.

**Related problem settings.**  A related setting that introduces a similar tradeoff to ours is when observations come at a cost (Stokey, 2008; Zhou et al., 2024). Our setting is distinct from this line of research as in our model, the cost of additional observations arises naturally from the loss of opportunity to intervene early, rather than being part of the modeling assumptions.

Our work is closely related to subsidy allocation in the presence of income shocks, as studied by Abebe et al. (2020). Their model captures a more general dynamic where individuals fail after experiencing potentially multiple shocks. Unlike Abebe et al. (2020), we do not assume a full information setting. In a similar vein, our proposed dynamic is also related to the dynamic models of opportunity allocation (Heidari & Kleinberg, 2021) and the dynamics of wealth across generations (Acharya et al., 2023).

Generally, the best estimates of the treatment effect are obtained when the allocation is randomized. This is often not the case when we need to learn while treating those in need (Wilder & Welle, 2024). The sample independence assumption is often violated in these settings (Shirali, 2022) and recent works have proposed various estimators to improve the power of estimators (Mate et al., 2023; Boehmer et al., 2024). Our model largely circumvents this complexity, as a simple ranking based on available observations is always Bayes optimal.

Our discussion is related to decision-focused learning (Mukhopadhyay & Vorobeychik, 2017; Wilder et al., 2019; Elmachtoub et al., 2020; Elmachtoub & Grigas, 2022) in the Operations Research community, where predictions are informed by their downstream applications. In our work, we employ a simplified observation model that allows us to consistently obtain a posterior distribution over hidden variables. This approach circumvents the challenges that could arise from inaccurate or biased predictions. There is also a direct connection between our Algorithm 1 and decision-focused learning. If we consider prediction as the ranking of individuals at *all* time points, then this algorithm effectively identifies the optimal prediction tailored for the subsequent allocation step.

Our over-time allocation is also related to multi-armed bandit problems with resource constraints in addition to reward (or utility) generation (Agrawal & Devanur, 2016; Slivkins et al., 2023). In particular, our model is most similar to the rotting bandit (Levine et al., 2017) where reward decreases over time. Unlike the standard bandit problems, in our problem observations are available from all

individuals and not only those treated. The exploration/exploitation tradeoff then lies in waiting for further information or treating those already estimated to be vulnerable.

**Prediction and policy problems.** Historically, policy planning has relied on aggregate data; however, the promise of improved resource allocation, reduced costs, and more preventative interventions has led to the widespread adoption of algorithmic systems on an individual basis in governments (Athey, 2017; Levy et al., 2021). Our work contributes to this discussion, as our insights have direct implications for policy planning in evolving social contexts.

While causal inference can inform policy-making, it is not always necessary (Kleinberg et al., 2015). Our framework falls under the category of prediction policy problems where an accurate ranking of individuals is sufficient for effective allocation. Related to this topic, Wang et al. (2024) raise concerns about the legitimacy of decision-making based on predictive optimization.

The debate surrounding risk assessment tools has largely centered around their inherently predictive nature. However, as emphasized by Barabas et al. (2018) in the context of the criminal justice system, the focus of risk assessment should be on guiding interventions rather than merely making predictions. Our research aligns with this perspective by studying prediction not as an isolated task but as an integral part of the resource allocation process.

**The long-lasting effect of interventions.** The Moving to Opportunity (MTO) experiment, sponsored by the U.S. Department of Housing and Urban Development, exemplifies an early intervention aimed at improving life outcomes by providing low-income families with children living in disadvantaged urban public housing the opportunity to relocate to less distressed private-market housing communities (de Souza Briggs et al., 2010; Ludwig et al., 2008; Gennetian et al., 2012; Ludwig et al., 2013; Chetty et al., 2016). The mixed findings of the MTO experiment across different age groups and the contrast between interim and long-term analyses highlight the crucial role that timing and the considered time horizon play in determining intervention's effect.

The MTO experiment also shows early interventions and environmental factors can have a long-lasting influence. In our model, we consider the extreme case of this when individuals subject to intervention are no longer vulnerable at any future time point. Hardt & Kim (2023) discuss how these long-lasting effects inform future predictions.

Shapiro (2004) argues that initial differences, rather than wage disparities, are the primary drivers of persistent inequality in the United States. Consistently, Derenoncourt (2022) show that while moving to areas with better economic opportunities theoretically provided improved prospects, local responses counteracted many of the potential benefits. Such complexities are all abstracted into the probability of failure in our model. While this abstraction helps us focus on specific aspects, we acknowledge that it does not capture the full range of dynamics in the real world.

## C   AN ILLUSTRATION OF TRADEOFFS IN ONE-TIME ALLOCATION

To further illustrate why the budget and inequality are key factors in determining the cost of waiting for accurate predictions in one-time allocation, we present the following example.

Suppose initially, $\mathcal{P}^1 = \mathrm{Beta}(1, G+1)$. This ensures that $\mathcal{P}^1$ is $G$-decaying according to Definition 4.1. Consider the problem of whether postponing the allocation from $t = 1$ to $t + 1 = 2$ is beneficial, or equivalently, whether $\Delta W^t := W^{t+1} - W^t > 0$. For simplicity of the illustration, we assume $\tilde{p} = p$ and fully effective treatments.

Denoting the set of active individuals at $t$ with $y^t = k$ by $\mathcal{A}_k^t$ and $N_k^t := |\mathcal{A}_k^t|$, a direct calculation gives

$$N_0^1 = N\,\mathbb{E}[1 - \tilde{p}] = N\frac{G+1}{G+2}, \qquad\qquad N_1^1 = N\,\mathbb{E}[\tilde{p}] = N\frac{1}{G+2},$$

$$N_1^2 = N\,\mathbb{E}[(1-p)(1-\tilde{p})\tilde{p}] = N\frac{2(G+1)}{(G+2)(G+3)}, \quad N_2^2 = N\,\mathbb{E}[(1-p)\tilde{p}^2] = N\frac{2}{(G+2)(G+3)}.$$

Note that $N_1^1 > N_2^2$ and $N_1^1 < N_1^2 + N_2^2$.

Lemma E.5 implies that the optimal allocation at $t$, should treat those who have higher $y^t$. Therefore, in our example, assuming $B < N_1^1$, the optimal allocation at $t = 1$ only treats those in $\mathcal{A}_1^1$. Similarly, at $t + 1 = 2$, it first treats those in $\mathcal{A}_2^2$ and then $\mathcal{A}_1^2$. So, defining $U_k^t := \mathbb{E}_{p \sim \mathcal{P}^t(\cdot|y^t=k)}[u^t(p)]$, for $B < N_1^1$, it is straightforward to obtain

$$\Delta W^t = \begin{cases} B\left(U_2^2 - U_1^1\right), & B \le N_2^2, \\ -B\left(U_1^1 - U_1^2\right) + N_2^2\left(U_2^2 - U_1^1\right), & B > N_2^2. \end{cases}$$

Lemma E.5 implies $U_1^1 \ge U_1^2$. Therefore, increasing the budget after $N_2^2$ can only decrease $\Delta W^t$ and favors earlier allocations. The effect of inequality is also well-captured in $U_2^2 - U_1^1$. A direct calculation shows $\mathrm{sign}(U_2^2 - U_1^1) = \mathrm{sign}\left((G+1)(T-4) - 6\right)$. Hence, $\Delta W^{>}0$ necessitates

$$G > \frac{6}{T-4} - 1.$$

In other words, higher inequality, reflected in smaller values of $G$, can render postponing allocation from $t$ to $t + 1$ unjustifiable.

We illustrate the effect of budget and inequality by simulating one-time allocation for $T = 6$ and $N = 10000$ in Fig. 3. Consistent with the theory, a high inequality corresponding to $G \le 2$ can make $W^{t+1} \le W^t$. Even when this is not the case, a high budget can still favor earlier allocation.

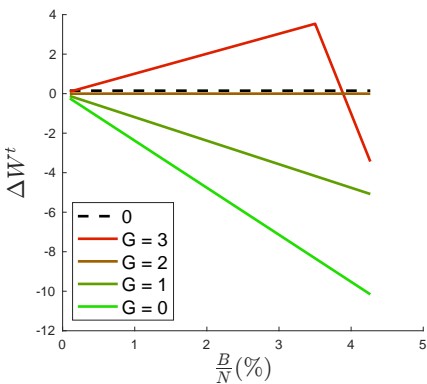

Figure 3: Illustration of the effect of the budget and inequality on one-time allocation.

# D  SUPPLEMENTARY ALGORITHMS

---

**Algorithm 2** Simulate trajectory

---

1: **inputs:**
2:   $t_0$ : starting time of simulation
3:   $\{N_k^{t_0}\}_{k=0}^{t_0}$ : number of individuals with $y^{t_0} = k$ for $k = 0, \ldots, t_0$
4:   $q : [T] \rightarrow \{0, 1, \ldots, T\}$: a valid non-decreasing sequence according to Theorem 5.1
5: **output:**
6:   $\{N_{q(t)}^t\}_{t=t_0}^T$: number of individuals reaching $y^t = q(t)$ who are not yet treated, for $t \geq t_0$
7: **function** SIMULATETRAJ($t_0, \{N_k^{t_0}\}_{k=0}^{t_0}, q(\cdot)$)
8:     **for** $t = t_0 + 1$ to $T$ **do**
9:         **for** $k = 0$ to $t$ **do**
            *Find the number of untreated individuals from the previous step:*
10:             $\overline{N}_k^{t-1} \leftarrow N_k^{t-1} \cdot \mathbb{1}\{k < q(t-1)\}, \quad \overline{N}_{k-1}^{t-1} \leftarrow N_{k-1}^{t-1} \cdot \mathbb{1}\{k-1 < q(t-1)\}$
            *Backup formula:*
                $N_k^t \leftarrow \overline{N}_k^{t-1} \cdot \mathbb{E}_{p \sim \mathcal{P}^{t-1}(\cdot | y^{t-1}=k)} \big[(1-p)(1-\tilde{p})\big]$
11:
                $\quad + \overline{N}_{k-1}^{t-1} \cdot \mathbb{E}_{p \sim \mathcal{P}^{t-1}(\cdot | y^{t-1}=k-1)} \big[(1-p)\,\tilde{p}\big]$
12:         **end for**
13:     **end for**
14:     **return** $\{N_{q(t)}^t\}_{t=t_0}^T$
15: **end function**

---

# E SUPPLEMENTARY STATEMENTS

## E.1 GENERAL STATEMENTS

**Lemma E.1.** *Suppose $f : [0, 1] \to \mathbb{R}$ is a non-decreasing function of $p$. Consider two probability density functions $\mathcal{P}$ and $\mathcal{Q}$ such that $\frac{\mathcal{P}(p)}{\mathcal{Q}(p)}$ is a non-decreasing continuous function of $p$. Then, we have $\mathbb{E}_P[f(p)] \geq \mathbb{E}_Q[g(p)]$. As a direct implication of this lemma, the sign of inequality flips if either $f$ or the density ratio is non-increasing.*

*Proof.* Define $\sigma(p) := \frac{\mathcal{P}(p)}{\mathcal{Q}(p)}$. Since $\int_0^1 \mathcal{Q}(p) \, dp = \int_0^1 \mathcal{Q}(p) \, \sigma(p) \, dp = 1$, and $\sigma$ is continuous, there should exist a critical value $p^*$ such that $\sigma(p) \geq 1$ for $p \geq p^*$, and $\sigma(p) \leq 1$ for $p < p^*$. Using this critical value to decompose the expectations and the fact that $f(\cdot)$ is non-decreasing, we obtain

$$
\begin{aligned}
\mathbb{E}_{\mathcal{P}}[f(p)] - \mathbb{E}_{\mathcal{Q}}[f(p)] &= \int_0^1 f(p) \, \mathcal{Q}(p) \, (\sigma(p) - 1) \, dp \\
&= \int_{p^*}^1 f(p) \, \mathcal{Q}(p) \, (\sigma(p) - 1) \, dp - \int_0^{p^*} f(p) \, \mathcal{Q}(p) \, (1 - \sigma(p)) \, dp \\
&\geq f(p^*) \int_{p^*}^1 \mathcal{Q}(p) \, (\sigma(p) - 1) \, dp - f(p^*) \int_0^{p^*} \mathcal{Q}(p) \, (1 - \sigma(p)) \, dp = 0 \, .
\end{aligned}
$$

$\square$

**Lemma E.2.** *Consider two non-increasing functions $a : \mathbb{R} \to [0, 1]$ and $b : \mathbb{R} \to [0, 1]$. If $\int_{-\infty}^{\infty} b(x) \, dx$ is finite and non-zero, the following inequality always holds:*

$$
\left( \int_{-\infty}^{\infty} b(x)^2 \, dx \right) \cdot \left( \int_{-\infty}^{\infty} a(x) \, b(x) \, dx \right) \geq \left( \int_{-\infty}^{\infty} a(x)^2 \, b(x)^2 \, dx \right) \cdot \left( \int_{-\infty}^{\infty} b(x) \, dx \right). \quad (9)
$$

*Proof.* Define the difference between the left-hand side and the right-hand side of the inequality given in Eq. (9) as $\Delta$. For simplicity, consider integrals as a discrete sum with a step size of $\delta$. Increasing the value of $a(x')$ would change the value of $\Delta$ by

$$
\frac{1}{\delta} \frac{d\Delta}{da(x')} = b(x') \cdot \int_{-\infty}^{\infty} b(x)^2 \, dx - 2a(x') \, b(x')^2 \cdot \int_{-\infty}^{\infty} b(x) \, dx \, .
$$

This implies that for any $x'$ such that $b(x') > 0$, increasing $a(x')$ will decrease $\Delta$ if and only if

$$
a(x') \, b(x') > \frac{1}{2} \frac{\int_{-\infty}^{\infty} b(x)^2 \, dx}{\int_{-\infty}^{\infty} b(x) \, dx} \, .
$$

Since both $a$ and $b$ are non-increasing non-negative functions, their multiplication is also a non-increasing non-negative function. Therefore, increasing $a(x')$ will decrease $\Delta$ if and only if $x'$ is larger than a critical value $x^*$. The non-increasing constraint on $a$ then implies that for a fixed $b$, the minimum value of $\Delta$ corresponds to a constant function $a(x) = a_0$. In this case,

$$
\Delta = \left( a_0 - a_0^2 \right) \cdot \left( \int_{-\infty}^{\infty} b(x)^2 \, dx \right) \cdot \left( \int_{-\infty}^{\infty} b(x) \, dx \right)
$$

For $a_0 \in [0, 1]$, the above equation is always greater than or equal to zero, which completes the proof. $\square$

## E.2 STATEMENTS ABOUT THE OPTIMALITY OF RANKING

**Lemma E.3** (Bayes optimal ranking). *Consider a statistical model $P = \{p_\theta : \theta \in \Theta = [a, b]\}$ that induces a family of continuous probability distributions over a sample space $\mathcal{X}$. Assume $P$ has a univariant sufficient statistics $T : \mathcal{X} \to \mathbb{R}$. Consider samples drawn independently from two probability distributions with distinct parameters: $X_1 \sim p_{\theta_1}, X_2 \sim p_{\theta_2}$. For a ranking function $\delta : \mathcal{X} \times \mathcal{X} \to \{-1, 1\}$, define the ranking loss as*

$$
\text{loss}((\theta_1, \theta_2); \delta(x_1, x_2)) := \mathbb{1}\{\delta(x_1, x_2)(\theta_2 - \theta_1) < 0\} \, .
$$

*Consider $\Theta_1$ and $\Theta_2$ independently drawn from a prior $\mathcal{P}$ over $\Theta$. If for any $\theta_2 \geq \theta_1$,*

$$T(x_2) \geq T(x_1) \iff p_{\theta_1}(x_1)\,p_{\theta_2}(x_2) \geq p_{\theta_1}(x_2)\,p_{\theta_2}(x_1)\,, \quad a.e.\,, \tag{10}$$

*then for any choice of $\mathcal{P}$ that has no point mass, the Bayes optimal ranking rule is $\delta^*(x_1, x_2) = \chi\{T(x_2) \geq T(x_1)\}$.*

*Proof.* For $\Theta_1$ and $\Theta_2$ independently drawn from $\mathcal{P}$, the Bayes risk of ranking is

$$R(\mathcal{P}^{\otimes 2}; \delta) = \mathbb{E}_{\substack{\Theta_1 \sim \mathcal{P} \\ \Theta_2 \sim \mathcal{P}}}\Big[\mathbb{E}_{\substack{X_1 \sim p_{\Theta_1} \\ X_2 \sim p_{\Theta_2}}}\big[\mathrm{loss}((\Theta_1, \Theta_2); \delta(X_1, X_2))\big]\Big].$$

The independence also allows us to decompose the posterior over $\Theta_1$ and $\Theta_2$ given $X_1 = x_1$ and $X_2 = x_2$ as $\mathcal{P}(\Theta_1 \mid x_1)\,\mathcal{P}(\Theta_2 \mid x_2)$. It is well-known that the minimizer of the Bayes risk is

$$\delta^*(x_1, x_2) = \underset{\delta(\cdot,\cdot)}{\arg\min}\, R(\mathcal{P}^{\otimes 2}; \delta) \in \underset{\delta \in \{-1,1\}}{\arg\min}\, \mathbb{E}_{\substack{\Theta_1 \sim \mathcal{P}(\cdot \mid x_1) \\ \Theta_2 \sim \mathcal{P}(\cdot \mid x_2)}}\big[\mathrm{loss}((\Theta_1, \Theta_2); \delta) \mid X_1 = x_1, X_2 = x_2\big].$$

Plugging the ranking loss into this, we can further simplify the conditional expectation and obtain

$$\begin{aligned}
\delta^*(x_1, x_2) &\in \underset{\delta \in \{-1,1\}}{\arg\min}\, \mathbb{E}_{\substack{\Theta_1 \sim \mathcal{P}(\cdot \mid x_1) \\ \Theta_2 \sim \mathcal{P}(\cdot \mid x_2)}}\big[\mathrm{loss}((\Theta_1, \Theta_2); \delta) \mid x_1, x_2\big] \\
&= \underset{\delta \in \{-1,1\}}{\arg\min}\, \frac{1+\delta}{2}\Pr(\Theta_1 > \Theta_2 \mid x_1, x_2) + \frac{1-\delta}{2}\Pr(\Theta_1 \leq \Theta_2 \mid x_1, x_2) \\
&= \underset{\delta \in \{-1,1\}}{\arg\min}\, \delta\big[\Pr(\Theta_1 > \Theta_2 \mid x_1, x_2) - \Pr(\Theta_1 \leq \Theta_2 \mid x_1, x_2)\big] \\
&= \mathrm{sign}\big(\Pr(\Theta_1 \leq \Theta_2 \mid x_1, x_2) - \Pr(\Theta_1 > \Theta_2 \mid x_1, x_2)\big).
\end{aligned}$$

Now, using a change of variable trick and the Bayes' rule, we have

$$\begin{aligned}
&\Pr(\Theta_1 \leq \Theta_2 \mid x_1, x_2) - \Pr(\Theta_1 > \Theta_2 \mid x_1, x_2) \\
&= \int_a^b \int_a^{\theta_2} \mathcal{P}(\theta_1 \mid x_1)\,\mathcal{P}(\theta_2 \mid x_2)\,\mathrm{d}\theta_1\,\mathrm{d}\theta_2 - \int_a^b \int_a^{\theta_1} \mathcal{P}(\theta_1 \mid x_1)\,\mathcal{P}(\theta_2 \mid x_2)\,\mathrm{d}\theta_2\,\mathrm{d}\theta_1 \\
&= \int_a^b \int_a^{\theta_2} \big[\mathcal{P}(\theta_1 \mid x_1)\,\mathcal{P}(\theta_2 \mid x_2) - \mathcal{P}(\theta_2 \mid x_1)\,\mathcal{P}(\theta_1 \mid x_2)\big]\,\mathrm{d}\theta_1\,\mathrm{d}\theta_2 \\
&= \int_a^b \int_a^{\theta_2} \frac{\mathcal{P}(\theta_1)\,\mathcal{P}(\theta_2)}{Z(x_1, x_2)}\big[p_{\theta_1}(x_1)\,p_{\theta_2}(x_2) - p_{\theta_1}(x_2)\,p_{\theta_2}(x_1)\big]\,\mathrm{d}\theta_1\,\mathrm{d}\theta_2\,,
\end{aligned}$$

where $Z(x_1, x_2)$ is the partition function. The integral bound enforces $\theta_2 \geq \theta_1$. Then if the condition of Eq. (10) holds, we can conclude

$$\mathrm{sign}\big(\Pr(\Theta_1 \leq \Theta_2 \mid x_1, x_2) - \Pr(\Theta_1 > \Theta_2 \mid x_1, x_2)\big) = \mathrm{sign}(T(x_2) - T(x_1))\,.$$

$\square$

**Proposition E.4.** *Consider the observation model $o \sim \mathrm{Ber}(\tilde{p})$, where $\tilde{p}(\cdot)$ is a non-decreasing function. Define $y^t = \sum_{t' \in [t]} o^t$. For any prior $\mathcal{P}$ over $p$ that has no point mass, ranking individuals based on their $y^t$ is Bayes optimal.*

*Proof.* At any time $t$, define the statistical model $P = \{p_\theta = (\mathrm{Ber}(\tilde{p}(\theta)))^{\otimes t} : \theta \in [0, 1]\}$ where we can think of the model parameter $\theta$ as the failure probability $p$. All the observations from individual $i$ until $t$ can be interpreted as a sample from a model in $P$: $X = [o^1, \ldots, o^t] \sim p_\theta$. Then, it is straightforward to see $y^t$ is a sufficient statistic for $P$ and $p_\theta(x) = \tilde{\theta}^{y^t}(1 - \tilde{\theta})^{t - y^t}$. The increasing property of $\tilde{p}$ also implies that $\theta_2 \geq \theta_1 \iff \tilde{\theta}_2 \geq \tilde{\theta}_1$.

For $\theta_2 \geq \theta_1$, plugging $p_\theta$ into the condition of Eq. (10) gives

$$p_{\theta_1}(x_1)\,p_{\theta_2}(x_2) \geq p_{\theta_1}(x_2)\,p_{\theta_2}(x_1) \iff \Big(\frac{\tilde{\theta}_2}{\tilde{\theta}_1}\frac{1 - \tilde{\theta}_1}{1 - \tilde{\theta}_2}\Big)^{y_2^t - y_1^t} \geq 1\,, \quad \text{a.e.}$$

Since for $1 > \tilde{\theta}_2 \geq \tilde{\theta}_1 > 0$, we have $\frac{\tilde{\theta}_2}{\tilde{\theta}_1}\frac{1-\tilde{\theta}_1}{1-\tilde{\theta}_2} \geq 1$, we can conclude

$$p_{\theta_1}(x_1)\,p_{\theta_2}(x_2) \geq p_{\theta_1}(x_2)\,p_{\theta_2}(x_1) \iff y_2^t \geq y_1^t\,, \quad \text{a.e.}$$

Therefore, $P$ meets the sufficient condition given in Eq. (10) of Lemma E.3, and ranking based on its sufficient statistic $y^t$ is Bayes optimal. $\square$

E.3 STATEMENTS ABOUT THE EFFECT OF ONE MORE OBSERVATION OR ONE MORE TIME STEP

**Lemma E.5** (Expected utility is monotone in the number of positive observations and time). *For any utility function $u^t(p)$ that is non-decreasing in $p$, we have*

$$\mathbb{E}_{k+1}^t[u^t(p)] \geq \mathbb{E}_k^t[u^t(p)],$$

*for every $k < t$. If the utility is also non-increasing in $t$, we have*

$$\mathbb{E}_k^t[u^t(p)] \geq \mathbb{E}_k^{t+1}[u^{t+1}(p)],$$

*for every $k \leq t$. Here, $\mathbb{E}_k^t$ denotes expectation with respect to $p \sim \mathcal{P}_k^t = \mathcal{P}^t(\cdot \mid y^t = k)$.*

*Proof.* We first prove the monotonicity in $k$. The likelihood $\mathrm{Pr}^t(y^t = k \mid p)$ has a closed-form of $\binom{t}{k}\tilde{p}^k(1-\tilde{p})^{t-k}$. Using this, we have

$$\frac{\mathcal{P}_{k+1}^t(p)}{\mathcal{P}_k^t(p)} \propto \frac{\mathrm{Pr}^t(y^t = k+1 \mid p)}{\mathrm{Pr}^t(y^t = k \mid p)} = \frac{\tilde{p}}{1-\tilde{p}}\Big(\frac{t-k}{k+1}\Big).$$

This is a non-decreasing continuous function of $\tilde{p}$ and, consequently, of $p$, for every $k < t$. Since the utility function is also non-decreasing in $p$, Lemma E.1 proves the monotonicity in $k$.

We next prove the monotonicity in $t$. Using the Bayes rule and the update rule of Eq. (7), we have

$$\frac{\mathcal{P}_k^t(p)}{\mathcal{P}_k^{t+1}(p)} \propto \frac{\mathcal{P}^t(p)\,\mathrm{Pr}^t(y^t = k \mid p)}{\mathcal{P}^{t+1}(p)\,\mathrm{Pr}^{t+1}(y^{t+1} = k \mid p)} \propto \Big(\frac{1}{1-p}\Big)\frac{\mathrm{Pr}^t(y^t = k \mid p)}{\mathrm{Pr}^{t+1}(y^{t+1} = k \mid p)}.$$

Plugging the closed-form expression of likelihoods into this, we obtain

$$\frac{\mathcal{P}_k^t(p)}{\mathcal{P}_k^{t+1}(p)} \propto \frac{1}{(1-p)(1-\tilde{p})}.$$

Again, this is a non-decreasing continuous function of $\tilde{p}$ and, consequently, of $p$, for every $k < t$. Since the utility function is also non-decreasing in $p$, Lemma E.1 implies $\mathbb{E}_k^t[u^t(p)] \geq \mathbb{E}_k^{t+1}[u^t(p)]$. Using the fact that the utility is also non-increasing in $t$, completes the proof. $\square$

**Lemma E.6** (A positive draw from $\mathrm{Ber}(p)$ is more informative than $\mathrm{Ber}(\tilde{p})$). *Consider the observation model $o \sim \mathrm{Ber}(\tilde{p})$, where $\tilde{p}(p)$ is a concave function with $\tilde{p}(0) = 0$ and $\tilde{p}(1) = 1$. Let $z \sim \mathrm{Ber}(p)$ be a random draw from an individual with failure probability $p$. For any utility function $u^t(p)$ non-increasing in $t$ and non-decreasing in $p$, we have*

$$\mathbb{E}_k^t[u^t(p) \mid z = 1] \geq \mathbb{E}_{k+1}^{t+1}[u^{t+1}(p)], \tag{11}$$

*for every $k \leq t$. Here, $\mathbb{E}_k^t[\cdot]$ denotes expectation with respect to $p \sim \mathcal{P}_k^t = \mathcal{P}^t(\cdot \mid y^t = k)$.*

*Proof.* Expanding the left-hand side of Eq. (11), we have

$$\mathbb{E}_k^t[u^t(p) \mid z = 1] = \frac{\mathbb{E}_k^t\big[\mathrm{Pr}(z = 1 \mid p) \cdot u^t(p)\big]}{\mathbb{E}_k^t\big[\mathrm{Pr}(z = 1 \mid p)\big]} = \frac{\mathbb{E}_k^t[p \cdot u^t(p)]}{\mathbb{E}_k^t[p]}.$$

On the other, expanding the right-hand side of Eq. (11) using the updating rule of $\mathcal{P}^{t+1}(p) \propto \mathcal{P}^t(p)(1-p)$ from Eq. (7), we have

$$\mathbb{E}_{k+1}^{t+1}[u^{t+1}(p)] = \frac{\mathbb{E}^{t+1}[\tilde{p}^{k+1}(1-\tilde{p})^{t-k} \cdot u^{t+1}(p)]}{\mathbb{E}^{t+1}[\tilde{p}^{k+1}(1-\tilde{p})^{t-k}]} = \frac{\mathbb{E}_k^t[(1-p)\tilde{p} \cdot u^{t+1}(p)]}{\mathbb{E}_k^t[(1-p)\tilde{p}]}.$$

Define $g(p) := (1-p)\tilde{p}$. We argue that $g(p)$ is concave on $[0,1]$. To see this, observe that $g''(p) = -2\tilde{p}' + (1-p)\tilde{p}''$. The concavity of $\tilde{p}$ implies $\tilde{p}'' \leq 0$. Then, given that $\tilde{p}(0) = 0$, $\tilde{p}(1) = 1$, and the range of $\tilde{p}$ is within $[0,1]$, it follows that $\tilde{p}' \leq 0$. Thus, we can conclude $g''(p) \leq 0$ for $p \in [0,1]$. A direct consequence of the concavity of $g$ is that $g'(p)(0-p) \geq g(0) - g(p) = -g(p)$. A straightforward integration then shows that for arbitrary $p_2 \geq p_1 > 0$,

$$\frac{p_2}{p_1} \geq \frac{g(p_2)}{g(p_1)}. \tag{12}$$

This inequality will allow us to further bound $\mathbb{E}_{k+1}^{t+1}[u^{t+1}(p)]$ as follows. Using $u^{t+1}(p) \leq u^t(p)$, the difference of the expanded sides of Eq. (11) can be bounded by

$$\frac{\mathbb{E}_k^t\big[(p_1 g(p_2) - p_2 g(p_1)) \cdot u^{t+1}(p_1)\big]}{\mathbb{E}_k^t[p_1 \cdot g(p_2)]} \,,$$

where $p_1$ and $p_2$ are two independent draws from $\mathcal{P}_k^t$. Using symmetry and the fact that the distribution has no point mass, we can write the numerator as

$$\mathbb{E}_k^t\big[(p_1 g(p_2) - p_2 g(p_1)) \cdot u^{t+1}(p_1) \cdot \big(\mathbb{1}\{p_2 \geq p_1\} + \mathbb{1}\{p_2 < p_1\}\big)\big]$$
$$= \mathbb{E}_k^t\big[(p_1 g(p_2) - p_2 g(p_1))(u^{t+1}(p_1) - u^{t+1}(p_2)) \cdot \mathbb{1}\{p_2 \geq p_1\}\big] \,.$$

Then the fact that utility is non-increasing in $p$ and Eq. (12) imply the numerator is non-negative. This completes the proof. $\qquad\square$

**Lemma E.7.** *Consider the observation model $o \sim \mathrm{Ber}(\tilde{p})$. Define $\mu_k^t := \mathbb{E}_k^t[p]$ and $\tilde{s}_k^t := \mathbb{E}_k^t[(1 - p)\tilde{p}]$, where $\mathbb{E}_k^t[\cdot]$ denotes expectation with respect to $p \sim \mathcal{P}_k^t = \mathcal{P}^t(\cdot \mid y^t = k)$. For any arbitrary function $f : [0, 1] \to \mathbb{R}$, the following identities hold:*

$$\mathbb{E}_k^{t+1}[f(p)] = \frac{\mathbb{E}_k^t[(1 - p)(1 - \tilde{p}) \cdot f(p)]}{1 - \mu_k^t - \tilde{s}_k^t} \,,$$

$$\mathbb{E}_{k+1}^{t+1}[f(p)] = \frac{1}{\tilde{s}_k^t} \mathbb{E}_k^t[(1 - p)\tilde{p} \cdot f(p)] \,.$$

*Proof.* Using the updating rule of $\mathcal{P}^{t+1}(p) \propto \mathcal{P}^t(p)(1 - p)$ from Eq. (7), we have

$$\mathbb{E}_k^{t+1}[f(p)] = \frac{\mathbb{E}^{t+1}[f(p) \cdot \tilde{p}^k(1 - \tilde{p})^{t+1-k}]}{\mathbb{E}^{t+1}[\tilde{p}^k(1 - \tilde{p})^{t+1-k}]} = \frac{\mathbb{E}^t[(1 - p)(1 - \tilde{p}) \cdot f(p) \cdot \tilde{p}^k(1 - \tilde{p})^{t-k}]}{\mathbb{E}^t[(1 - p)(1 - \tilde{p}) \cdot \tilde{p}^k(1 - \tilde{p})^{t-k}]}$$
$$= \frac{\mathbb{E}_k^t[(1 - p)(1 - \tilde{p}) \cdot f(p)]}{1 - \mu_k^t - \tilde{s}_k^t} \,.$$

Using the same techniques, we also obtain

$$\mathbb{E}_{k+1}^{t+1}[f(p)] = \frac{\mathbb{E}^{t+1}[f(p) \cdot \tilde{p}^{k+1}(1 - \tilde{p})^{t-k}]}{\mathbb{E}^{t+1}[\tilde{p}^{k+1}(1 - \tilde{p})^{t-k}]} = \frac{\mathbb{E}^t[(1 - p)\tilde{p} \cdot f(p) \cdot \tilde{p}^k(1 - \tilde{p})^{t-k}]}{\mathbb{E}^t[(1 - p)\tilde{p} \cdot \tilde{p}^k(1 - \tilde{p})^{t-k}]}$$
$$= \frac{1}{\tilde{s}_k^t} \mathbb{E}_k^t[(1 - p)\tilde{p} \cdot f(p)] \,.$$

$\qquad\square$

**Lemma E.8** (Bounding the effect of one more positive observation on expected utility). *Consider the observation model $o \sim \mathrm{Ber}(\tilde{p})$, where $\tilde{p}(p)$ is a concave function with $\tilde{p}(0) = 0$ and $\tilde{p}(1) = 1$. Define $U_k^t := \mathbb{E}_k^t[u^t(p)]$ and $\mu_k^t := \mathbb{E}_k^t[p]$ as the expected utility and the expected failure probability of an active individual at time $t$ who has $k$ positive observations. For any concave $(\lambda_1, \lambda_2)$-decaying $L_u(t)$-Lipschitz utility function $u^t$, we have*

$$U_{k+1}^{t+1} - U_k^{t+1} \leq L_u(t + 1) \cdot (\lambda_2 - U_k^{t+1}) \cdot \frac{\mu_{k+1}^{t+1} - \mu_k^{t+1}}{1 - \mu_k^{t+1}} \,.$$

*Proof.* Denote the cumulative distribution function corresponding to $\mathcal{P}_k^t$ by $\mathcal{F}_k^t$. The concavity of $\tilde{p}$ and its boundary values imply $\tilde{p}(p)$ is non-decreasing in $p$. Then, a straightforward argument shows $\mathcal{F}_k^{t+1}(p) \leq \mathcal{F}_{k+1}^{t+1}(p)$. Let $\pi : [0, 1] \to [0, 1]$ be the optimal transport map from $\mathcal{P}_k^{t+1}$ to $\mathcal{P}_{k+1}^{t+1}$. The definition of $\pi$ implies

$$U_{k+1}^{t+1} - U_k^{t+1} = \mathbb{E}_k^{t+1}\big[u^{t+1}(\pi(p)) - u^{t+1}(p)\big] \,.$$

Using the concavity of the utility function, we can upper bound the above by

$$U_{k+1}^{t+1} - U_k^{t+1} \leq \mathbb{E}_k^{t+1}\big[(u^{t+1})'(p) \cdot (\pi(p) - p)\big] \,.$$

Our observation that $\mathcal{F}_k^{t+1}(p) \le \mathcal{F}_{k+1}^{t+1}(p)$ implies that $\pi(p) \ge p$. In other words, the optimal map must always shift the mass to the right. Furthermore, since $\frac{\mathcal{P}_{k+1}^{t+1}}{\mathcal{P}_k^{t+1}} \propto \frac{\tilde{p}}{1-\tilde{p}}$ is an increasing function of $\tilde{p}$ (and therefore $p$), it follows that $\pi(p) - p$ is non-decreasing in $p$. The concavity of utility also implies that $(u^{t+1})'(p)$ is non-increasing. Then, Chebyshev's sum inequality allows us to bound $U_{k+1}^{t+1} - U_k^{t+1}$ as the product of two terms:

$$U_{k+1}^{t+1} - U_k^{t+1} \le \mathbb{E}_k^{t+1}[(u^{t+1})'(p)] \cdot \mathbb{E}_k^{t+1}[\pi(p) - p] = \mathbb{E}_k^{t+1}[(u^{t+1})'(p)] \cdot (\mu_{k+1}^{t+1} - \mu_k^{t+1}). \quad (13)$$

Since the utility is concave, $(u^{t+1})'$ is non-increasing. This enables us to apply Chebyshev's sum inequality, allowing us to write

$$\mathbb{E}_k^{t+1}[(u^{t+1})'(p) \cdot (1-p)] \ge \mathbb{E}_k^{t+1}[(u^{t+1})'(p)] \cdot (1 - \mu_k^{t+1}),$$

which gives an upper bound on $\mathbb{E}_k^{t+1}[(u^{t+1})'(p)]$. Now, since the utility function is $(\lambda_1, \lambda_2)$-decaying, we have

$$
\begin{aligned}
\mathbb{E}_k^{t+1}[(u^{t+1})'(p)] &\le \frac{\mathbb{E}_k^{t+1}[(u^{t+1})'(p) \cdot (1-p)]}{1 - \mu_k^{t+1}} \\
&\le \frac{\mathbb{E}^{t+1}[L_u(t+1) \cdot (\lambda_2 - u^{t+1}(p)) \cdot \tilde{p}^k(1-\tilde{p})^{t+1-k}]}{\mathbb{E}^{t+1}[\tilde{p}^k(1-\tilde{p})^{t+1-k}] \cdot (1 - \mu_k^{t+1})} \\
&= L_u(t+1) \frac{\lambda_2 - U_k^{t+1}}{1 - \mu_k^{t+1}}.
\end{aligned}
$$

Plugging this into Eq. (13) completes the proof. $\qquad\square$

**Lemma E.9** (Bounding the effect of one more observation and one more time step on expected failure probability). *Consider the observation model $o \sim \mathrm{Ber}(\tilde{p})$, where $\tilde{p}(p) = 1 - (1-p)^\gamma$ and $\gamma > 1$. Suppose the initial distribution $\mathcal{P}^1$ is G-decaying. For any $k < t$, the following inequalities hold:*

$$\mu_{k+1}^t \ge \mu_{k+1}^{t+1}, \qquad\qquad\qquad\qquad \text{(effect of one more time step)}$$

$$\mu_{k+1}^{t+1} - \mu_k^{t+1} \le \mu_{k+1}^{t+1}(1 - \mu_{k+1}^{t+1})\left(\frac{1}{k+1} + \frac{2 + \gamma + \gamma^{-1} + G}{\gamma(t-k) + t + 1}\right). \quad \text{(effect of one more observation)}$$

*Proof.* The proof of the effect of one additional time step is straightforward: Starting from the definition of $\mu_{k+1}^t$ and using the updating rule $\mathcal{P}^{t+1}(p) \propto \mathcal{P}^t(p)(1-p)$ from Eq. (7), we have

$$\mu_{k+1}^t = \frac{\mathbb{E}^t[p \cdot \tilde{p}^{k+1}(1-\tilde{p})^{t-k-1}]}{\mathbb{E}^t[\tilde{p}^{k+1}(1-\tilde{p})^{t-k-1}]} = \frac{\mathbb{E}_{k+1}^{t+1}[p \cdot (1-p)^{-1}(1-\tilde{p})^{-1}]}{\mathbb{E}_{k+1}^{t+1}[(1-p)^{-1}(1-\tilde{p})^{-1}]}.$$

Then, since $(1-\tilde{p})$ is non-increasing in $p$, a direct application of Lemma E.1 gives $\mu_k^{t+1} \ge \mu_{k+1}^{t+1}$.

In the second part, we prove the effect of one more observation. We start by writing $\mu_k^{t+1}$ as

$$\mu_k^{t+1} = \frac{\mathbb{E}^{t+1}[p \cdot \tilde{p}^k(1-\tilde{p})^{t+1-k}]}{\mathbb{E}^{t+1}[\tilde{p}^k(1-\tilde{p})^{t+1-k}]} = \frac{\mathbb{E}^{t+1}[p \cdot \frac{\mathrm{d}}{\mathrm{d}p}(\tilde{p}^{k+1}) \cdot (1-\tilde{p})^{t-k} \cdot (1-p)]}{\mathbb{E}^{t+1}[\frac{\mathrm{d}}{\mathrm{d}p}(\tilde{p}^{k+1}) \cdot (1-\tilde{p})^{t-k} \cdot (1-p)]}. \quad (14)$$

Here, we used the identity $\tilde{p}' = \gamma \frac{1-\tilde{p}}{1-p}$. Using integration by parts, we can write the numerator as

$$
\begin{aligned}
\mathbb{E}^{t+1}[p \cdot \frac{\mathrm{d}}{\mathrm{d}\tilde{p}}(\tilde{p}^{k+1}) \cdot (1-\tilde{p})^{t-k} \cdot (1-p)] =\ & p \cdot \tilde{p}^{k+1}(1-\tilde{p})^{t-k} \cdot \mathcal{P}^{t+1}(p)(1-p)\Big|_0^1 \\
& - \int_0^1 \tilde{p}^{k+1}(1-\tilde{p})^{t-k} \cdot \mathcal{P}^{t+1}(p)(1-p)\,\mathrm{d}p \\
& + \gamma(t-k)\int_0^1 p \cdot \tilde{p}^{k+1}(1-\tilde{p})^{t-k} \cdot \mathcal{P}^{t+1}(p)\,\mathrm{d}p \\
& - \int_0^1 p \cdot \tilde{p}^{k+1}(1-\tilde{p})^{t-k} \cdot \frac{\mathrm{d}}{\mathrm{d}p}\big(\mathcal{P}^{t+1}(p)(1-p)\big)\,\mathrm{d}p.
\end{aligned}
$$

Here, we again applied the identity $\tilde{p}' = \gamma \frac{1-\tilde{p}}{1-p}$ to arrive at the third term. The first term is zero. We next simplify and bound the last term. Using the updating rule $\mathcal{P}^{t+1}(p) \propto \mathcal{P}^t(p)(1-p)$ from Eq. (7) we can write

$$\mathcal{P}^{t+1}(p)(1-p) = \frac{\mathcal{P}^1(p)(1-p)^{t+1}}{Z^{t+1}} \,,$$

where $Z^{t+1}$ is a normalizing constant. Taking the derivative with respect to $p$ and simplifying equations, we obtain

$$\frac{\mathrm{d}}{\mathrm{d}p}\big(\mathcal{P}^{t+1}(p)(1-p)\big) = -(t+1)\mathcal{P}^{t+1}(p) + \frac{(\mathcal{P}^1)'(p)(1-p)^t}{Z^{t+1}} \,.$$

Since $\mathcal{P}^1$ is $G$-decaying, we have

$$-(t+1+G)\mathcal{P}^{t+1}(p) \leq \frac{\mathrm{d}}{\mathrm{d}p}\big(\mathcal{P}^{t+1}(p)(1-p)\big) \leq -(t+1)\mathcal{P}^{t+1}(p) \,.$$

Using these bounds in the expansion via integration by parts and simplifying the integrals as expectations, we can impose the following bounds:

$$(\gamma\,(t-k)+t+2)\,\mu_{k+1}^{t+1}-1 \leq \mathbb{E}_{k+1}^{t+1}[p \cdot \frac{\mathrm{d}}{\mathrm{d}\tilde{p}}(\tilde{p}^{k+1}) \cdot (1-\tilde{p})^{t-k} \cdot (1-p)] \leq (\gamma\,(t-k)+t+G+2)\,\mu_{k+1}^{t+1}-1 \,.$$

Using similar arguments, we can also derive the following bounds:

$$\gamma\,(t-k)+t+1 \leq \mathbb{E}_{k+1}^{t+1}[\frac{\mathrm{d}}{\mathrm{d}\tilde{p}}(\tilde{p}^{k+1}) \cdot (1-\tilde{p})^{t-k} \cdot (1-p)] \leq \gamma\,(t-k)+t+G+1 \,.$$

Plugging these bounds into Eq. (14), we obtain

$$\frac{(\gamma\,(t-k)+t+2)\,\mu_{k+1}^{t+1}-1}{\gamma\,(t-k)+t+G+1} \leq \mu_k^{t+1} \leq \frac{(\gamma\,(t-k)+t+G+2)\,\mu_{k+1}^{t+1}-1}{\gamma\,(t-k)+t+1} \,. \tag{15}$$

Using the lower bound from Eq. (15), after a straightforward calculation, we obtain

$$\frac{\mu_{k+1}^{t+1}-\mu_k^{t+1}}{\mu_{k+1}^{t+1}\,(1-\mu_{k+1}^{t+1})} \leq \frac{1}{\gamma\,(t-k)+t+G+1}\left[\frac{1}{\mu_{k+1}^{t+1}}+\frac{G}{1-\mu_{k+1}^{t+1}}\right] \,. \tag{16}$$

One can verify that the maximum of the terms inside the brackets occurs only when $\mu_{k+1}^{t+1}$ reaches its smallest or largest values. Here, we only present the case where $\mu_{k+1}^{t+1}$ takes its smallest value, but a similar bound will hold when it takes its largest value. Therefore, the last missing piece of the proof is a lower bound on $\mu_{k+1}^{t+1}$. Using the upper bound from Eq. (15), we have

$$\mu_{k+1}^{t+1} \geq \mu_k^{t+1} + \frac{1-(G+1)\mu_{k+1}^{t+1}}{\gamma\,(t-k)+t+1} \,.$$

Repetitively applying the above operation yields

$$\mu_{k+1}^{t+1} \geq \mu_0^{t+1} + \sum_{k'=1}^{k+1} \frac{1}{\gamma\,(t-k')+\gamma+t+1} - (G+1)\sum_{k'=1}^{k+1} \frac{\mu_{k'}^{t+1}}{\gamma\,(t-k')+\gamma+t+1} \,.$$

An implication of Lemma E.1 is $\mu_{k'}^{t+1} \leq \mu_{k'+1}^{t+1}$. We also know $\mu_0^{t+1} \geq 0$. Using these, we can obtain the lower bound

$$\mu_{k+1}^{t+1} \geq \frac{S}{1+(G+1)S} \,,$$

where $S = \sum_{k'=1}^{k+1}(\gamma\,(t-k')+\gamma+t+1)^{-1}$. Using the naive bound $S \geq \frac{k+1}{\gamma t+t+1}$, we can further lower bound $\mu_{k+1}^{t+1}$ as

$$\mu_{k+1}^{t+1} \geq \frac{k+1}{\gamma t+t+1+(k+1)(G+1)} \,.$$

Plugging this into Eq. (16) gives

$$\frac{\mu_{k+1}^{t+1}-\mu_k^{t+1}}{\mu_{k+1}^{t+1}\,(1-\mu_{k+1}^{t+1})} \leq \frac{\gamma t+t+1+(k+1)(G+1)}{\gamma\,(t-k)+t+G+1}\left[\frac{1}{k+1}+\frac{1}{\gamma t+t+1+(k+1)G}\right] \,.$$

Without further ado, using $k \leq t$ and simplifying the equations complete the proof:

$$\frac{\mu_{k+1}^{t+1}-\mu_k^{t+1}}{\mu_{k+1}^{t+1}\,(1-\mu_{k+1}^{t+1})} \leq \frac{1}{k+1}+\frac{2+\gamma+\gamma^{-1}+G}{\gamma\,(t-k)+t+1} \,.$$

$\square$

### E.4 OTHER STATEMENTS

**Proposition E.10.** *Suppose the distribution $\mathcal{P}$ over $p$ is $G$-decaying according to Definition 4.1. This guarantees*

$$\mu := \mathbb{E}[p] \geq \underline{\mu} := \frac{1}{2+G}\,,$$

$$\sigma^2 := \text{Var}[p] \geq \underline{\sigma}^2 := \frac{2\mu}{3+G} - \mu^2\,.$$

*Note that $\frac{d\underline{\mu}}{dG} < 0$ and $\frac{\partial\underline{\sigma}^2}{\partial G} < 0$. The above inequalities are tight for $\mathcal{P} = \text{Beta}(1, 1+G)$.*

*Proof.* Using $G$-decaying property of $\mathcal{P}$ and integrating by parts, we obtain

$$\begin{aligned}
\mu = \int_0^1 p\,\mathcal{P}(p)\,dp &\geq -\frac{1}{G}\int_0^1 (1-p)p\,\frac{d\mathcal{P}}{dp}\,dp \\
&= -\frac{1}{G}(1-p)p\mathcal{P}(p)\Big|_0^1 + \frac{1}{G}\mathbb{E}\big[\frac{d(1-p)p}{dp}\big] \\
&= \frac{1-2\mu}{G}\,.
\end{aligned}$$

Rearranging the terms proves $\mu \geq \underline{\mu} := \frac{1}{2+G}$.

Similarly, using the $G$-decaying property of $\mathcal{P}$ and integrating by parts again, we have

$$\begin{aligned}
\sigma^2 + \mu^2 = \mathbb{E}[p^2] = \int_0^1 p^2\,\mathcal{P}(p)\,dp &\geq -\frac{1}{G}\int_0^1 (1-p)p^2\frac{d\mathcal{P}}{dp}\,dp \\
&= -\frac{1}{G}(1-p)p^2\mathcal{P}(p)\Big|_0^1 + \frac{1}{G}\mathbb{E}\big[\frac{d(1-p)p^2}{dp}\big] \\
&= \frac{2\mu - 3\mathbb{E}[p^2]}{G}\,.
\end{aligned}$$

Rearranging the terms, we obtain

$$\mathbb{E}[p^2] \geq \frac{2\mu}{3+G}\,.$$

This implies $\sigma^2 \geq \underline{\sigma}^2 := \frac{2\mu}{3+G} - \mu^2$. $\qquad\square$

**Proposition E.11.** *The following notions of utility fit into our definition of $(\lambda_1, \lambda_2)$-decaying utilities:*

1. *If the treatment is fully effective, the utility function is $(1, 1)$-decaying.*

2. *If the treatment succeeds with probability $c$ and otherwise fails the individual, as in the case of a risky medical procedure, the utility function is $(c, c)$-decaying.*

3. *If the treatment succeeds with probability $\alpha$ and otherwise has no effect, the utility function is $(\alpha, 1)$-decaying.*

4. *If the treatment reduced failure probability from $p$ to $p/\gamma$ for $\gamma > 1$, the utility function is $\big((1 - \gamma^{-1})^{T-t}, 1\big)$-decaying.*

*Proof.* The proof follows by plugging $u^t(p)$ of each case into Definition 4.4:

1. In case of fully effective treatments, $u^t(p) = 1 - (1-p)^{T-t}$. The decrease in utility over time is

$$u^t(p) - u^{t+1}(p)\,(1-p) = p\,.$$

Therefore, $\lambda_1 \leq 1$. The increase in utility with $p$ is

$$(1-p)\frac{(u^t)'(p)}{(u^t)'(0)} + u^t(p) = (1-p)^{T-t} + u^t(p) = 1\,.$$

So, $\lambda_2 \geq 1$. Putting these together, $u^t(p)$ is $(1, 1)$-decaying.

2. When treatment succeeds with probability $c$ and fails otherwise, $u^t(p) = c - (1-p)^{T-t}$. The decrease in utility over time is

$$u^t(p) - u^{t+1}(p)\,(1-p) = c\,p\,.$$

Therefore, $\lambda_1 \leq c$. The increase in utility with $p$ is

$$(1-p)\frac{(u^t)'(p)}{(u^t)'(0)} + u^t(p) = (1-p)^{T-t} + u^t(p) = c\,.$$

Hence, $\lambda_2 \geq c$. Putting these together, $u^t(p)$ is $(c, c)$-decaying.

3. When treatment succeeds with probability $\alpha$ and has no effect otherwise, $u^t(p) = \alpha + (1 - \alpha)(1-p)^{T-t} - (1-p)^{T-t} = \alpha - \alpha\,(1-p)^{T-t}$. The decrease in utility over time is

$$u^t(p) - u^{t+1}(p)\,(1-p) = \alpha - \alpha\,(1-p) = \alpha\,p\,.$$

So, $\lambda_1 \leq \alpha$. The increase in utility with $p$ is bounded by

$$(1-p)\frac{(u^t)'(p)}{(u^t)'(0)} + u^t(p) = (1-p)^{T-t} + u^t(p) = (1-\alpha)(1-p)^{T-t} + \alpha \leq 1\,.$$

Hence, $\lambda_2 \geq 1$. Putting these together, $u^t(p)$ is $(\alpha, 1)$-decaying.

4. In case treatment reduces $p$ to $p/\gamma$, we have $u^t(p) = (1 - p/\gamma)^{T-t} - (1-p)^{T-t}$. The decrease in utility over time is bounded by

$$
\begin{aligned}
u^t(p) - u^{t+1}(p)\,(1-p) &= (1-p/\gamma)^{T-t} - (1-p)(1-p/\gamma)^{T-t-1} \\
&= (1-p/\gamma)^{T-t-1}(1-1/\gamma)\,p \geq (1-1/\gamma)^{T-t}\,p\,.
\end{aligned}
$$

Hence, $\lambda_1 \leq (1 - \gamma^{-1})^{T-t}$. The increase in utility with $p$ is bounded by

$$
\begin{aligned}
&(1-p)\frac{(u^t)'(p)}{(u^t)'(0)} + u^t(p) \\
&= (1-p)\frac{(1-p)^{T-t-1} - \frac{1}{\gamma}(1-p/\gamma)^{T-t-1}}{1 - 1/\gamma} + (1-p/\gamma)^{T-t} - (1-p)^{T-t} \\
&= \frac{\frac{1}{\gamma}(1-p)^{T-t} - (1-p/\gamma)^{T-t-1}\left[1/\gamma - p/\gamma - (1-1/\gamma)(1-p/\gamma)\right]}{1 - 1/\gamma} \\
&= \frac{\frac{1}{\gamma}(1-p)^{T-t} - \frac{1}{\gamma}(1-p/\gamma)^{T-t}}{1 - 1/\gamma} + (1-p/\gamma)^{T-t-1} \leq 1\,.
\end{aligned}
$$

So, $\lambda_2 \geq 1$. Putting these together, $u^t(p)$ is $\left((1 - \gamma^{-1})^{T-t}, 1\right)$-decaying.

$\square$

**Lemma E.12.** *Consider the observation model $o \sim \mathrm{Ber}(\tilde{p})$, where $\tilde{p}(p) = 1 - (1-p)^\gamma$ and $\gamma > 1$. Suppose the initial distribution $\mathcal{P}^1$ is non-increasing. Let $l^t$ be the smallest $y_i^t$ such that individual $i$ would be treated at $t$ given a budget of $B$ to be spent at $t$. We have*

$$l^t \leq (\gamma + 1) \cdot \ln\left(\frac{N}{B}\right).$$

*Proof.* For notational brevity, let $k = l^t$. Define the complementary cumulative distribution function (CCDF) of a binomial random variable by

$$b(k, t) := \sum_{k'=k+1}^{t} \binom{t}{k'} \tilde{p}^{k'} (1 - \tilde{p})^{t-k'}\,.$$

Note that $b(\cdot, \cdot)$ implicitly depends on $p$, which we have omitted from the notation for brevity. The following identity will be proved to be useful:

$$
\begin{aligned}
\frac{\mathrm{d}b(k,t)}{\mathrm{d}\tilde{p}} &= \sum_{k'=k+1}^{t} \binom{t}{k'} k' \cdot \tilde{p}^{k'-1}(1-\tilde{p})^{t-k'} - \sum_{k'=k+1}^{t-1} \binom{t}{k'}(t-k') \cdot \tilde{p}^{k'}(1-\tilde{p})^{t-k'-1} \\
&= t \cdot \tilde{p}^{t-1} + \sum_{k'=k+1}^{t-1} \binom{t}{k'}(k'-t\tilde{p}) \cdot \tilde{p}^{k'-1}(1-\tilde{p})^{t-k'-1} \\
&= t \cdot \tilde{p}^{t-1} + t \cdot \sum_{k'=k+1}^{t-1} \binom{t-1}{k'-1} \tilde{p}^{k'-1}(1-\tilde{p})^{t-k'-1} - t \cdot \sum_{k'=k+1}^{t-1} \binom{t}{k'} \tilde{p}^{k'}(1-\tilde{p})^{t-k'-1} \\
&= t \cdot \tilde{p}^{t-1} + \frac{t}{1-\tilde{p}} \cdot \big(b(k-1,t-1) - \tilde{p}^{t-1}\big) - \frac{t}{1-\tilde{p}}\big(b(k,t) - \tilde{p}^t\big) \\
&= \frac{t}{1-\tilde{p}} \cdot \big(b(k-1,t-1) - b(k,t)\big).
\end{aligned}
$$

Intuitively, the above identity relates the derivative of the CCDF with respect to $\tilde{p}$ to its finite difference across time. The CCDF is also related to the number of individuals with $y^t > k$. Denoting the number of individuals with $y^t = k'$ by $N_{k'}^t$, we can write:

$$
\sum_{k'=k+1}^{t} N_{k'}^t = N^t \cdot \mathbb{E}^t[b(k,t)].
$$

Using the two identities presented above, we have

$$
\begin{aligned}
\sum_{k'=k}^{t-1} N_{k'}^{t-1} - \sum_{k'=k+1}^{t} N_{k'}^t &= N^{t-1} \cdot \mathbb{E}^{t-1}[b(k-1,t-1)] - N^t \cdot \mathbb{E}^t[b(k,t)] \\
&= N^{t-1} \cdot \mathbb{E}^{t-1}[b(k,t)] - N^t \cdot \mathbb{E}^t[b(k,t)] + \frac{1}{t} N^{t-1} \cdot \mathbb{E}^{t-1}[(1-\tilde{p})\frac{\mathrm{d}b(k,t)}{\mathrm{d}\tilde{p}}].
\end{aligned}
$$

Now, applying the updating rule $\mathcal{P}^{t+1}(p) \propto \mathcal{P}^t(p)(1-p)$ from Eq. (7), we obtain

$$
\begin{aligned}
\sum_{k'=k}^{t-1} N_{k'}^{t-1} - \sum_{k'=k+1}^{t} N_{k'}^t &= N^t \cdot \mathbb{E}^t[\frac{p}{1-p}b(k,t)] + \frac{1}{t} N^t \cdot \mathbb{E}^t[(\frac{1-\tilde{p}}{1-p})\frac{\mathrm{d}b(k,t)}{\mathrm{d}\tilde{p}}] \\
&\geq \frac{1}{t} N^t \cdot \mathbb{E}^t[(1-\tilde{p})\frac{\mathrm{d}b(k,t)}{\mathrm{d}\tilde{p}}].
\end{aligned} \tag{17}
$$

Using integration by parts, we can expand the expectation:

$$
\mathbb{E}^t[(1-\tilde{p})\frac{\mathrm{d}b(k,t)}{\mathrm{d}\tilde{p}}] = \widetilde{\mathcal{P}}^t(\tilde{p})(1-\tilde{p}) \cdot b(k,t)\Big|_0^1 - \int_0^1 b(k,t) \cdot \frac{\mathrm{d}}{\mathrm{d}\tilde{p}}\big(\widetilde{\mathcal{P}}^t(\tilde{p})(1-\tilde{p})\big)\, \mathrm{d}\tilde{p}.
$$

Here, $\widetilde{\mathcal{P}}$ denotes the distribution over $\tilde{p}$. For $k < t$, the first term above is zero. To bound the second term, note that $\widetilde{\mathcal{P}}^t(\tilde{p}) = \frac{\mathcal{P}^t(p)}{\tilde{p}'}$. Then, using the identity $\tilde{p}' = \gamma\big(\frac{1-\tilde{p}}{1-p}\big)$, we have

$$
\frac{\mathrm{d}}{\mathrm{d}\tilde{p}}\big(\widetilde{\mathcal{P}}^t(\tilde{p})(1-\tilde{p})\big) = \frac{1}{\gamma\,\tilde{p}'}\frac{\mathrm{d}}{\mathrm{d}p}\big(\mathcal{P}^t(p)(1-p)\big) \leq -\frac{t}{\gamma}\widetilde{\mathcal{P}}^t(\tilde{p}).
$$

We used $(\mathcal{P}^t)'(p) \leq 0$ to arrive at the above inequality. Plugging this into Eq. (17) and doing a straightforward calculation, we obtain

$$
\big(\frac{\gamma}{\gamma+1}\big) \sum_{k'=k}^{t-1} N_{k'}^{t-1} \geq \sum_{k'=k+1}^{t} N_{k'}^t.
$$

By repetitively applying such inequalities, we have

$$
\big(\frac{\gamma}{\gamma+1}\big)^k \sum_{k'=0}^{t-k} N_{k'}^t \geq \sum_{k'=k}^{t} N_{k'}^t.
$$

Since the last individual who is treated at $t$ has $y^t = k$, the right-hand side is lower bounded by $B$. The sum in the left-hand side is also bounded by the total number of initial individuals. These yield the following bound on $k$:

$$k \leq \frac{\ln(\frac{N}{B})}{\ln(1 + \gamma^{-1})}.$$

Using $\ln(1 + x) \geq \frac{x}{1+x}$ completes the proof. □

## F    MISSING PROOFS

*Proof of Theorem 3.1.* Using the step function approximation, the exponential multiplier in Eq. (8) is only non-zero if

$$\frac{|\tilde{p}_j - \tilde{p}_i|}{\tilde{\sigma}_{ij}} \leq \sqrt{\frac{2 \ln(1/\alpha)}{t}}.$$

This implies the following lower bound on $\Delta R^t$:

$$\Delta R^t \geq \sqrt{\frac{1}{4\pi \ln(1/\alpha)}} \cdot \mathbb{E}_{i,j}^t \left[ \frac{(1 - p_i)(1 - p_j)}{(1 - \mu^t)^2} - 1 \Big| \frac{|\tilde{p}_j - \tilde{p}_i|}{\tilde{\sigma}_{ij}} \leq \sqrt{\frac{2 \ln(1/\alpha)}{t}} \right] - \sqrt{\frac{\ln(1/\alpha)}{4\pi t^2}}.$$

Suppose $\tilde{p}^{-1}(\cdot)$ is $L^{-1}$-Lipschitz continuous. Using this and $\tilde{\sigma}_{ij}^2 \leq 1/2$, we can further bound $\Delta R^t$ by

$$\Delta R^t \geq \sqrt{\frac{1}{4\pi \ln(1/\alpha)}} \cdot \mathbb{E}^t \left[ \frac{(1 - p)(1 - p - \frac{L^{-1}}{1-2\epsilon}\sqrt{\frac{2 \ln(1/\alpha)}{t}})}{(1 - \mu^t)^2} - 1 \right] - \sqrt{\frac{\ln(1/\alpha)}{4\pi t^2}}$$

$$= \sqrt{\frac{1}{4\pi \ln(1/\alpha)}} \cdot \frac{\mathrm{Var}^t[p] - (1 - \mu^t)\frac{L^{-1}}{1-2\epsilon}\sqrt{\frac{2 \ln(1/\alpha)}{t}}}{(1 - \mu^t)^2} - \sqrt{\frac{\ln(1/\alpha)}{4\pi t^2}}.$$

The ranking improves over time when $\Delta R^t < 0$. In this case, it is necessary for the lower bound presented above to be negative as well. This completes the proof with $C_{\text{approx.}} = \ln(1/\alpha)$. □

*Proof of Theorem 4.3.* We build on the more general results of Theorem 4.5. First, observe from Proposition E.11 that the fully effective treatment corresponds to a $(1, 1)$-decaying utility function. Define $t_{\text{warm-up}} := (\gamma + 1) \ln(\frac{N}{B})$. Plugging $\lambda_1 = \lambda_2 = 1$ and $(u^{t+1})'(0) = T - t - 1$ into Theorem 4.5, we have

$$T - (t + 1) \geq \frac{t - t_{\text{warm-up}}}{1 + C \frac{t_{\text{warm-up}}+1}{2t - t_{\text{warm-up}}+1}}. \tag{18}$$

Here, we summarized $2 + \gamma + \gamma^{-1} + G$ into $C$. Consider two cases depending on whether $t$ is less than or larger than $t_c := (T + t_{\text{warm-up}} - 1)/2$. When $t \geq t_c$, we can relax the right-hand side of Eq. (18) by

$$T - (t + 1) \geq \frac{t - t_{\text{warm-up}}}{1 + C \frac{k+1}{T}}.$$

Simplifying this bound through a straightforward calculation, we get

$$t + 1 \leq \frac{T + (t_{\text{warm-up}} + 1)(C + 1)}{2 + (t_{\text{warm-up}} + 1)\frac{C}{T}}$$

$$= \frac{T}{2} + \frac{(t_{\text{warm-up}} + 1)(\frac{C}{2} + 1)}{2 + (t_{\text{warm-up}} + 1)\frac{C}{T}}$$

$$\leq \frac{T}{2} + (t_{\text{warm-up}} + 1)(\frac{1}{2} + \frac{C}{4}).$$

Therefore, we can conclude, either $t < t_c$ or the above bound should hold. Since the above bound is larger than $t_c$, this is the looser bound. Setting this to $t^*$ completes the proof. □

*Proof of Theorem 4.5.* We use the following shorthands and notation throughout the proof. Let $N_k^t \coloneqq |\mathcal{A}_k^t|$ where $\mathcal{A}_k^t$ is the set of active individuals at $t$ with $y^t = k$. In the limit of many individuals, $N_k^t = N^t \cdot \mathbb{E}^t[\binom{t}{k}\tilde{p}^k(1-\tilde{p})^{t-k}]$. Denote the posterior over $p$ given $y^t = k$ by $\mathcal{P}_k^t$. The Bayes' rule implies $\mathcal{P}_k^t(p) \propto \mathcal{P}^t(p)\tilde{p}^k(1-\tilde{p})^{t-k}$. Denote the expectation over individuals in $\mathcal{A}_k^t$ by $\mathbb{E}_k^t[\cdot]$. In the limit of many individuals, $\mathbb{E}_k^t[\cdot] = \mathbb{E}_{p\sim\mathcal{P}_k^t, o\sim\text{Ber}(\tilde{p})}[\cdot]$. We use the shorthand $U_k^t$ to denote $\mathbb{E}_k^t[u^t(p)]$ and $\mu_k^t$ to denote $\mathbb{E}_k^t[p]$ and $\tilde{s}_k^t$ to denote $\mathbb{E}_k^t[(1-p)\tilde{p}]$.

Assuming the prior over $p$ has no point mass, Proposition E.4 implies ranking individuals in descending order of $y^t$ is optimal. Therefore, given a budget of $B$, the optimal planner sorts individuals in descending order of $y^t$ and allocates to the top $B$. Let $l^t$ be the *lowest* $y^t$ of an individual that has received the allocation.

Suppose at time $t$, we have $l^t = k$. Then $B > N_t^t$ implies $k < t$. We argue that postponing the allocation from time $t$ to time $t+1$ can only increase overall utility if $l^{t+1}$ is either $k$ or $k+1$:

- We first rule out $l^{t+1} > k+1$. If $l^{t+1} > k+1$, only individuals with $y^{t+1} > k+1$ can be treated. This will be a subset of the individuals that could be treated at $t$. In particular, individuals with $y^t = k$ will not be eligible at $t+1$. Therefore, the whole budget has not been spent which violates the optimality of allocation at $t+1$.

- We next rule out $l^{t+1} < k$. The key observation is Lemma E.6, which states that the expected utility of an individual with $y^t = k$ who fails at the next step is higher than that of an individual with $y^{t+1} = k+1$. In other words, failing provides a stronger signal than $o$ regarding the individual's likelihood of failure. Formally, when $l^{t+1} < k$, everyone who was eligible for treatment at time $t$ and survived to $t+1$ remains eligible. The individuals who failed during this transition are replaced by new individuals with $y^{t+1} \leq k+1$. In the best case, the expected utility from each new individual is $U_{k+1}^{t+1}$. However, Lemma E.6 suggests that the expected utility of those who failed and were replaced is at least $U_{k+1}^{t+1}$. Therefore, postponing allocation cannot be justified.

In the remainder of the proof, we derive an upper bound on $\Delta W^t \coloneqq W^{t+1} - W^t$ when $l^{t+1}$ is either $k$ or $k+1$. We will then further simplify this bound to obtain sufficient conditions for $\Delta W^t \leq 0$.

- We first examine the case where $l^{t+1} = k$. Suppose at time $t$, we are able to treat $\Delta^t$ individuals from $\mathcal{A}_k^t$. This number changes to $\Delta^{t+1}$ at time $t+1$. The change in total utility after postponing allocation for one step can be written as

$$\Delta W^t = W^{t+1} - W^t = \left(\Delta^{t+1}U_k^{t+1} + \sum_{k'=k+1}^{t+1} N_{k'}^{t+1}U_{k'}^{t+1}\right) - \left(\Delta^t U_k^t + \sum_{k'=k+1}^{t} N_{k'}^t U_{k'}^t\right).$$

From $\mathcal{A}_k^t$, a ratio of $\mu_k^t$ fail and a ratio of $\tilde{s}_k^t$ proceed to the next step while revealing one more positive observation. This allows us to relate $N_{k'}^{t+1}$ with $N_{k'}^t$ and $N_{k'-1}^t$ for $k' \geq 1$:

$$N_{k'}^{t+1} = N_{k'}^t(1 - \mu_{k'}^t - \tilde{s}_{k'}^t) + N_{k'-1}^t \tilde{s}_{k'-1}^t. \tag{19}$$

Plugging this into $W^{t+1}$, we obtain

$$W^{t+1} = \Delta^{t+1}U_k^{t+1} + \sum_{k'=k+1}^{t} N_{k'}^t(1 - \mu_{k'}^t - \tilde{s}_{k'}^t)U_{k'}^{t+1} + \sum_{k'=k+1}^{t+1} N_{k'-1}^t \tilde{s}_{k'-1}^t U_{k'}^{t+1}$$

$$= \Delta^{t+1}U_k^{t+1} + N_k^t \tilde{s}_k^t U_{k+1}^{t+1} + \sum_{k'=k+1}^{t} N_{k'}^t\left[(1 - \mu_{k'}^t - \tilde{s}_{k'}^t)U_{k'}^{t+1} + \tilde{s}_{k'}^t U_{k'+1}^{t+1}\right].$$

The summand above can be significantly simplified. Using Lemma E.7 with $f = u^{t+1}$, we have

$$(1 - \mu_{k'}^t - \tilde{s}_{k'}^t)U_{k'}^{t+1} + \tilde{s}_{k'}^t U_{k'+1}^{t+1} = \mathbb{E}_{k'}^t[(1-p) \cdot u^{t+1}(p)].$$

Since the utility is $(\lambda_1, \lambda_2)$-decaying, we can upper bound the above by

$$\mathbb{E}_{k'}^t[(1-p) \cdot u^{t+1}(p)] \leq \mathbb{E}_{k'}^t[u^t(p) - \lambda_1 p] = U_{k'}^t - \lambda_1 \mu_{k'}^t. \tag{20}$$

Plugging this into $W^{t+1}$ allows us to upper bound $\Delta W^t$:

$$\Delta W^t \le \Delta^{t+1} U_k^{t+1} - \Delta^t U_k^t + N_k^t \tilde{s}_k^t U_{k+1}^{t+1} - \lambda_1 \sum_{k'=k+1}^{t} N_{k'}^t \mu_{k'}^t . \tag{21}$$

Spending a fixed budget requires

$$\Delta^{t+1} = \Delta^t + \sum_{k'=k+1}^{t} N_{k'}^t - \sum_{k'=k+1}^{t+1} N_{k'}^{t+1} .$$

Using Eq. (19) and performing a straightforward calculation, we can simplify $\Delta^{t+1}$ as

$$\Delta^{t+1} = \Delta^t - N_k^t \tilde{s}_k^t + \sum_{k'=k+1}^{t} N_{k'}^t \mu_{k'}^t . \tag{22}$$

Plugging this into Eq. (21), we obtain

$$\Delta W^t \le -\Delta^t (U_k^t - U_k^{t+1}) + N_k^t \tilde{s}_k^t (U_{k+1}^{t+1} - U_k^{t+1}) - \big( \sum_{k=k+1}^{t} N_{k'}^t \mu_{k'}^t \big)(\lambda_1 - U_k^{t+1}) . \tag{23}$$

For a utility function non-increasing in time, we have $U_k^{t+1} \le \mathbb{E}_k^{t+1}[u^t(p)]$. Since $\frac{\mathcal{P}_k^{t+1}(p)}{\mathcal{P}_k^t(p)} \propto (1-p)(1-\tilde{p})$ is continuous and decreasing in $p$, for a utility function non-decreasing in $p$, Lemma E.1 implies $\mathbb{E}_k^{t+1}[u^t(p)] \le \mathbb{E}_k^t[u^t(p)] = U_k^t$. Therefore, we can conclude the upper bound of Eq. (23) is decreasing in $\Delta^t$. Thus, we can further upper bound $\Delta W^t$ by finding an upper bound on $\Delta^t$. Since $l^{t+1} = k$, it is necessary to have $\Delta^{t+1} \ge 0$. Using Eq. (22), this will impose an upper bound on $\Delta^t$:

$$\Delta^t \ge N_k^t \tilde{s}_k^t - \sum_{k'=k+1}^{t} N_{k'}^t \mu_{k'}^t .$$

Plugging this into Eq. (23), we have

$$\Delta W^t \le N_k^t \tilde{s}_k^t (U_{k+1}^{t+1} - U_k^t) - \big( \sum_{k=k+1}^{t} N_{k'}^t \mu_{k'}^t \big)(\lambda_1 - U_k^t) . \tag{24}$$

- We next consider the case of $l^{t+1} = k + 1$. We follow steps similar to those in the previous case. The change in total utility after postponing allocation for one step is

$$\Delta W^t = W^{t+1} - W^t = \Big( (\Delta^{t+1} - N_{k+1}^{t+1}) U_{k+1}^{t+1} + \sum_{k'=k+1}^{t+1} N_{k'}^{t+1} U_{k'}^{t+1} \Big)$$
$$- \Big( \Delta^t U_k^t + \sum_{k'=k+1}^{t} N_{k'}^t U_{k'}^t \Big) .$$

Plugging $N_{k'}^{t+1}$ expansion from Eq. (19) into $W^{t+1}$, we obtain

$$W^{t+1} = (\Delta^{t+1} - N_{k+1}^{t+1}) U_{k+1}^{t+1} + \sum_{k'=k+1}^{t} N_{k'}^t (1 - \mu_{k'}^t - \tilde{s}_{k'}^t) U_{k'}^{t+1} + \sum_{k'=k+1}^{t+1} N_{k'-1}^t \tilde{s}_{k'-1}^t U_{k'}^{t+1}$$
$$= (\Delta^{t+1} - N_{k+1}^{t+1}) U_{k+1}^{t+1} + N_k^t \tilde{s}_k^t U_{k+1}^{t+1} + \sum_{k'=k+1}^{t} N_{k'}^t \big[ (1 - \mu_{k'}^t - \tilde{s}_{k'}^t) U_{k'}^{t+1} + \tilde{s}_{k'}^t U_{k'+1}^{t+1} \big] .$$

As we showed in the previous case, the summand above can be bounded by $U_{k'}^t - \lambda_1 \mu_{k'}^t$. Plugging this into $W^{t+1}$ allows us to upper bound $\Delta W^t$:

$$\Delta W^t \le (\Delta^{t+1} - N_{k+1}^{t+1}) U_{k+1}^{t+1} - \Delta^t U_k^t + N_k^t \tilde{s}_k^t U_{k+1}^{t+1} - \lambda_1 \sum_{k'=k+1}^{t} N_{k'}^t \mu_{k'}^t . \tag{25}$$

Spending a fixed budget requires

$$\Delta^t = (\Delta^{t+1} - N_{k+1}^{t+1}) + \sum_{k'=k+1}^{t+1} N_{k'}^{t+1} - \sum_{k'=k+1}^{t} N_{k'}^{t}.$$

Using Eq. (19) and performing a straightforward calculation, we can simplify $\Delta^{t+1}$ as

$$\Delta^t = (\Delta^{t+1} - N_{k+1}^{t+1}) + N_k^t \tilde{s}_k^t - \sum_{k'=k+1}^{t} N_{k'}^t \mu_{k'}^t.$$

Plugging this into Eq. (25), we obtain

$$\Delta W^t \leq (N_{k+1}^{t+1} - \Delta^{t+1})(U_k^t - U_{k+1}^{t+1}) + N_k^t \tilde{s}_k^t (U_{k+1}^{t+1} - U_k^t) - \big( \sum_{k=k+1}^{t} N_{k'}^t \mu_{k'}^t \big)(\lambda_1 - U_k^t).$$

We argue postponing allocation is only justified if $U_{k+1}^{t+1} > U_k^t$. Otherwise, the eligible individuals with the lowest $y^t$ at the previous step already have higher expected utility than the newly eligible individuals at $t+1$. This implies that the above upper bound is increasing in $\Delta^{t+1}$. The largest $\Delta^{t+1}$ happens when $\Delta^{t+1}$ approaches $N_{k+1}^{t+1}$. The reader can already verify this will lead to the same bound as the previous case (Eq. (24)).

Before proceeding to analyze the upper bound of Eq. (24), we make one more observation. Recall that in Eq. (20) we used $(\lambda_1, \lambda_2)$-decaying property of the utility function. However, Lemma E.6 implies another complementary bound of this quantity:

$$\mathbb{E}_{k'}^t[(1 - p) \cdot u^{t+1}(p)] \leq U_{k'}^t - \mathbb{E}_{k'}^t[p \cdot u^{t+1}(p)] \leq U_{k'}^t - U_{k'+1}^{t+1} \mu_{k'}^t.$$

Comparing this with the bound in Eq. (20), we observe that any result involving $\lambda_1$ can always be updated by substituting $\lambda_1$ with $U_{k+1}^{t+1}$. Such an update to Eq. (24) results in

$$\Delta W^t \leq (N_k^t \tilde{s}_k^t - \sum_{k=k+1}^{t} N_{k'}^t \mu_{k'}^t)(U_{k+1}^{t+1} - U_k^t).$$

Recall that deferring allocation requires $U_{k+1}^{t+1} > U_k^t$. The above equation further implies that $N_k^t \tilde{s}_k^t \geq \sum_{k=k+1}^{t} N_{k'}^t \mu_{k'}^t$ is also necessary. Now, looking back at Eq. (24), we can conclude that under these necessary conditions, the upper bound is decreasing in $U_k^t$. Hence, decreasing $U_k^t$ to $U_k^{t+1}$ can only increase the bound:

$$\Delta W^t \leq N_k^t \tilde{s}_k^t (U_{k+1}^{t+1} - U_k^{t+1}) - \big( \sum_{k=k+1}^{t} N_{k'}^t \mu_{k'}^t \big)(\lambda_1 - U_k^{t+1}). \tag{26}$$

We will use this inequality as a basis for the remainder of the proof.

The following identity will be useful:

$$N_k^t \tilde{s}_k^t = \binom{t}{k} N^t \cdot \mathbb{E}^t[(1 - p)\tilde{p} \cdot \tilde{p}^k (1 - \tilde{p})^{t-k}]$$

$$= \binom{t}{k} N^{t+1} \cdot \mathbb{E}^{t+1}[\tilde{p}^{k+1}(1 - \tilde{p})^{t-k}] = \big(\frac{k+1}{t+1}\big) N_{k+1}^{t+1}. \tag{27}$$

Here, we used the updating rule $\mathcal{P}^{t+1}(p) = \mathcal{P}^t(p)\big(\frac{1-p}{1-\mu^t}\big)$ from Eq. (7). Eq. (27) implies that individuals with $y^t = k$ who survive to the next step while showing one more positive observation, form $\big(\frac{k+1}{t+1}\big)$ of $N_{k+1}^{t+1}$. So we can conclude that the rest of $N_{k+1}^{t+1}$ comes from $N_{k+1}^t$. Therefore, we have

$$N_{k+1}^t \geq \big(\frac{t-k}{t+1}\big) N_{k+1}^{t+1}. \tag{28}$$

Plugging this and Eq. (27) into Eq. (26), we obtain

$$\Delta W^t \leq N_{k+1}^{t+1} \big(\frac{k+1}{t+1}\big) \Big[(U_{k+1}^{t+1} - U_k^{t+1}) - \big(\frac{t-k}{k+1}\big)\mu_{k+1}^t (\lambda_1 - U_k^{t+1})\Big]. \tag{29}$$

In the rest of the proof, we consider two cases where the utility function is either $(\lambda_1, \lambda_2)$-decaying or just $\lambda_1$-decaying.

**The case of $(\lambda_1, \lambda_2)$-decaying utility.** The difference $(U_{k+1}^{t+1} - U_k^{t+1})$ in Eq. (29) captures the effect of one more observation on the expected utility. For a $(\lambda_1, \lambda_2)$-decaying utility, Lemma E.8 upper bounds this effect by

$$U_{k+1}^{t+1} - U_k^{t+1} \le (u^{t+1})'(0) \cdot (\lambda_2 - U_k^{t+1}) \cdot \frac{\mu_{k+1}^{t+1} - \mu_k^{t+1}}{1 - \mu_k^{t+1}} \,.$$

Here, we used $(u^{t+1})'(0)$ in place of the Lipschitz constant of $u^{t+1}$ in the lemma because the utility function is concave. Plugging this into Eq. (29) and using the assumption that $\lambda_1 \ge \lambda_2$, we obtain

$$\Delta W^t \le C\Big[(u^{t+1})'(0) - \frac{\lambda_1}{\lambda_2}\big(\frac{t-k}{k+1}\big)\frac{\mu_{k+1}^t(1 - \mu_k^{t+1})}{\mu_{k+1}^{t+1} - \mu_k^{t+1}}\Big] \,.$$

Here, we summarized all the terms multiplying before the bracket as a constant $C$, since these terms do not affect the sign of $\Delta W^t$. It is straightforward to verify $\mu_{k+1}^t \ge \mu_{k+1}^{t+1}$ and $\mu_k^{t+1} \le \mu_{k+1}^{t+1}$. Then, for $\gamma > 1$, Lemma E.9 yields

$$\Delta W^t \le C\Big[(u^{t+1})'(0) - \big(\frac{\lambda_1}{\lambda_2}\big)\frac{t-k}{1 + (2 + \gamma + \gamma^{-1} + G)\frac{k+1}{2t-k+1}}\Big] \,. \tag{30}$$

The upper bound of Eq. (30) is increasing in $k$. Therefore, the last missing piece of the proof is an upper bound on $k$. Lemma E.12 provides such an upper bound and completes this part of the proof.

**The case of $\lambda_1$-decaying utility.** As we discussed above, decreasing $U_k^t$ in Eq. (24) can only increase the bound. Setting $U_k^t$ then gives

$$\Delta W^t \le N_k^t \tilde{s}_k^t U_{k+1}^{t+1} - \Big(\sum_{k=k+1}^t N_{k'}^t \mu_{k'}^t\Big)\lambda_1 \,.$$

Plugging Eq. (28) into this, we obtain

$$\Delta W^t \le N_{k+1}^{t+1}\big(\frac{k+1}{t+1}\big)\Big[U_{k+1}^{t+1} - \big(\frac{t-k}{k+1}\big)\mu_{k+1}^t\lambda_1\Big] \,. \tag{31}$$

Concavity of the utility function and its zero value at $p = 0$ imply

$$U_{k+1}^{t+1} \le (u^{t+1})'(0) \cdot \mu_{k+1}^{t+1} \,.$$

Plugging this into Eq. (31), we have

$$\Delta W^t \le C\Big[(u^{t+1})'(0) - \big(\frac{t-k}{k+1}\big)\lambda_1\Big] \,,$$

where all the factors outside of the brackets are summarized in $C$. Using the upper bound on $k$ from Lemma E.12 completes this part of the proof.

**Extending to all future times.** As the final remark of the proof, note that the upper bounds provided in all cases are shrinking with $t$. Therefore, if $\Delta W^t \le 0$, then postponing allocation to any future time cannot be justified. □

*Proof of Theorem 5.1.* Consider $N$ individuals initially at time $t = 1$. Denote the subset of $\mathcal{A}^t$ with $y^t = k$ by $\mathcal{A}_k^t$. Define $n_k^t := |\mathcal{A}_k^t|/N$. Denote the set of individuals treated at $t$ by $\mathcal{I}^t$. Excluding $\mathcal{I}^t$ from $\mathcal{A}_k^t$, denote the remaining by $\bar{\mathcal{A}}_k^t := \mathcal{A}^t \setminus \mathcal{I}^t$, and define $\bar{n}_k^t := |\bar{\mathcal{A}}_k^t|/N$. In the limit of $N \to \infty$, we can treat $n_k^t$ and $\bar{n}_k^t$ as continuous variables taking any value in $[0, 1]$.

*Step* 1. The following property of the problem dynamics allows us to infer whether $\mathcal{A}_k^t$ is empty or not based on the previous step.

*Lemma* F.1. *Defining $\bar{n}_{-1}^t = 0$, at any time $t$ and for any $k$, we have*

$$n_k^{t+1} > 0 \iff \bar{n}_k^t > 0 \text{ or } \bar{n}_{k-1}^t > 0 \,.$$

See proof on page 35.

*Step* 2. Consider two active individuals $i$ and $j$ at time $t$. If $y_i^t > y_j^t$, for any utility function non-decreasing in $p$, Lemma E.5 implies treating $i$ yields more utility than $j$ in expectation. Therefore, the optimal allocation at any point should not target individuals with lower $y^t$ while there are active individuals with a higher $y^t$.

*Step* 3. We show that, except for the first time step, at each time $t$ on the optimal path, the treated individuals at $t$ should have similar values of $y^t$.

*Lemma* F.2. *For $\mathcal{A}^t \neq \emptyset$ and $t \geq 2$ on the optimal path, for any $i, j \in \mathcal{I}^t$, we should have $y_i^t = y_j^t = \max\{i' \mid i' \in \mathcal{A}^t\}$.*

See proof on page 35.

*Step* 4. When an individual $i$ with $y_i^t = k \geq 1$ is treated at $t$, not only is $\mathcal{A}_k^t$ non-empty, but $\mathcal{A}_{k-1}^t$ is also non-empty.

*Lemma* F.3. *For $k \geq 1$, if there exists $i \in \mathcal{I}^t$ on the optimal path such that $y_i^t = k$, then $n_{k-1}^t > 0$.*

See proof on page 35.

*Step* 5. Using the structure imposed on the optimal solution in the previous steps, we next restrict $\mathcal{I}^{t+1}$ based on $\mathcal{I}^t$.

*Lemma* F.4. *Suppose $\mathcal{A}^t \neq \emptyset$ and let $k = \max\{y_i^t \mid i \in \mathcal{A}^t\}$. On the optimal path,*

- *If $\mathcal{I}^t = \mathcal{A}_k^t$, either $\mathcal{I}^{t+1} \subseteq \mathcal{A}_k^{t+1}$ or $\mathcal{I}^{t+1} = \mathcal{A}_{k+1}^{t+1} = \emptyset$.*

- *If $\mathcal{I}^t \subset \mathcal{A}_k^t$, either $\mathcal{I}^{t+1} \subseteq \mathcal{A}_{k+1}^{t+1}$ or $\mathcal{I}^{t+1} = \mathcal{A}_{k+2}^{t+1} = \emptyset$.*

See proof on page 35.

*Step* 6. Except for one time step, at every $t$ on the optimal path, either $\mathcal{I}^t = \emptyset$ or $\mathcal{I}^t$ treats everyone with $y^t \geq k$ for some $k$. If there were two time steps $t$ and $t'$ violating this, because of the linearity of the expected utility in $|\mathcal{I}^t|$ and $|\mathcal{I}^{t'}|$, optimally, one would become zero or treat everyone above a cutoff. This and Lemma F.4 complete the proof.

$\square$

*Proof of Lemma 5.2.* We first iterate over two of the three unspecified parameters, $\hat{t}$ and $q(\cdot)$ in Theorem 5.1. There are $T$ possibilities for $\hat{t}$. A valid sequence $q(\cdot)$ can then be determined by specifying $q(1)$ and a binary sequence of length $T - 1$. The $t^{\text{th}}$ binary value in this sequence determines whether $q(t + 1) - q(t)$ will be: 0 or 1 if $t \neq \hat{t}$, or 1 or 2 if $t = \hat{t}$. Since the maximum $y^t$ at each time $t$ is $t$, it is straightforward to verify that for any sequence that starts with $q(1) > 2$, there exists another sequence with $q(1) \leq 2$ that treats similar individuals. Therefore, there are effectively only three choices for $q(1)$. Together, iterating over these parameters requires $3\,T \cdot 2^{T-1}$ steps.

Given $\hat{t}$ and a valid sequence $q(\cdot)$, there remains to determine the allocation at $\hat{t}$ based on the available budget. Let $\rho$ denote the proportion of $\mathcal{A}_{q(\hat{t})}^{\hat{t}}$ who are not treated. One can verify that for a fixed $\hat{t}$ and $q(\cdot)$, both the spending and the expected total utility are linear in $\rho$. Therefore, to find $\rho$, we only need to simulate an allocation for two distinct values of $\rho$. We then use these two points to identify the linear relationships and determine the optimal $\rho$ under the budget constraint. In Algorithm 1, we perform this by simulating the trajectories for $\rho = 0$ and $\rho = 1$, which takes $O(T^2)$ operations. So, overall, we require $O(T^3 \cdot 2^{T-1})$ operations to iterate over all rollouts. $\square$

*Proof of Theorem 5.3.* While iterating the search space through Lemma 5.2, we can store the number of individuals from $\mathcal{A}_k^t$ treated at each $t$ and $k$. Then there remains to calculate $U_k^t := \mathbb{E}_{p \sim \mathcal{P}^t(\cdot \mid y^t = k)}[u^t(p)]$. Assuming the expectation under posterior distribution can be calculated efficiently in constant steps, we can calculate the total utility of an instance in $O(T^2)$. Since this is happening in the same loop of iteration as Lemma 5.2, it does not add to the asymptotic complexity of the algorithm. $\square$

*Proof of Lemma F.1.* For $k \geq 1$, $\mathcal{A}_k^{t+1}$ will consist of those in $\bar{\mathcal{A}}_k^t$ who survive and have $o^{t+1} = 0$, or those in $\bar{\mathcal{A}}_{k-1}^t$ who survive and have $o^{t+1} = 1$:

$$n_k^{t+1} = \bar{n}_k^t \, \mathbb{E}_k^t \left[ (1-p)(1-\tilde{p}) \right] + \bar{n}_{k-1}^t \, \mathbb{E}_{k-1}^t \left[ (1-p)\tilde{p} \right].$$

Here, $\mathbb{E}_k^t$ denotes expectation with respect to $p \sim \mathcal{P}_k^t = \mathcal{P}^t(\cdot \mid y^t = k)$. We implicitly used the fact that the targeting cannot distinguish people with the same $y^t$. The same update rule works for $k = 0$ if we set $\bar{n}_{-1}^t = 0$. One can also verify that since the prior over $p$ has no point mass, the expectations above are non-zero. This completes the proof. $\qquad\square$

*Proof of Lemma F.2.* The proof is by contradiction with optimality and has multiple steps:

- Let $k = \max \{ y_i^t \mid i \in \mathcal{A}^t \}$. Since $n_{k+1}^t = 0$, Lemma F.1 requires $\bar{n}_{k+1}^{t-1} = \bar{n}_k^{t-1} = 0$. On the other hand, when $n_k^t > 0$, Lemma F.1 requires either $\bar{n}_k^{t-1}$ or $\bar{n}_{k-1}^{t-1}$ to be non-zero. Since we just argued $\bar{n}_k^{t-1} = 0$, it is required to have $\bar{n}_{k-1}^{t-1} > 0$.

- Since $\bar{n}_{k-1}^{t-1} > 0$, Lemma F.1 implies $n_{k-1}^t > 0$. Then Step 2 requires that if $\mathcal{I}^t$ contains individuals with different $y^t$, there should be two individuals $i, j \in \mathcal{I}^t$ such that $y_i^t = k$ and $y_j^t = k-1$.

- Consider the following tie-breaking when treating individuals with a similar $y^t$: Assign a random priority value $z_i^1 \in [0, 1)$ to each individual $i$ in the initial pool. At time $t$, update the priority value by $z_i^t = z_i^{t-1} + o_i^t \, 2^{t-1}$. If any two individuals are at a tie to receive the treatment, choose the one with the lowest priority value. This tie-breaking does not change the optimality of an allocation rule.

- So far we showed $\bar{n}_{k-1}^{t-1} > 0$, i.e., there exist some individuals in $\mathcal{A}_{k-1}^{t-1}$ left untreated at $t-1$, but there exists $j \in \mathcal{A}_{k-1}^t$ who is treated at $t$. We argue this is suboptimal as the budget to treat $j$ could have been spent earlier to treat those in $\mathcal{A}_{k-1}^{t-1}$, yielding higher utility. To see this, consider a counterfactual allocation rule that treats one more individual from $\mathcal{A}_{k-1}^{t-1}$ at $t-1$, to be referred to as individual $j'$. This individual has either failed or made it to $t$. If failed, Lemma E.6 implies that the counterfactual treatment could be more effective. If $j'$ is active at $t$, she will either have $y_{j'}^t = k$ or $y_{j'}^t = k-1$. If $y_{j'}^t = k$, then she is treated with others in $\mathcal{A}_k^t$. If $y_{j'}^t = k-1$, still $j'$ is treated. This is because $j'$ maintains the lowest priority value among $\mathcal{A}_{k-1}^t$ by the construction of the priority values. Since $j'$ is treated in any case if she makes it to $t$, she could be treated earlier at $t-1$. This could yield a higher or similar utility as $u^t$ is non-increasing in $t$. This contradiction shows $y_i^t = y_j^t = k$.

  $\qquad\square$

*Proof of Lemma F.3.* The proof is obvious for $t = 1$ since $k$ can only be 1 and unless the budget is excessively large to treat everyone, we have $n_0^1 > 0$. For $t \geq 2$, Lemma F.2 requires $n_k^t > 0$ and $n_{k+1}^t = 0$. Then Lemma F.1 implies $\bar{n}_k^{t-1} = 0$ and $\bar{n}_{k-1}^{t-1} > 0$, so $n_{k-1}^t > 0$. $\qquad\square$

*Proof of Lemma F.4.* We first prove the first part the lemma. If $\mathcal{I}^t = \mathcal{A}_k^t$, there are two possibilities for $k$. If $k = 0$, then $\bar{\mathcal{A}}_k^t = \emptyset$ and Step 2 implies no one is left untreated at $t$. So, $\mathcal{I}^{t+1} = \mathcal{A}_{k+1}^{t+1} = \mathcal{A}^{t+1} = \emptyset$. If $k \geq 1$, then Lemma F.3 implies $n_{k-1}^t > 0$. Then $\mathcal{I}_k^t = \mathcal{A}_k^t$ implies $\bar{n}_{k-1}^t > 0$ and $\bar{n}_k^t = 0$. Applying Lemma F.1 gives $n_k^{t+1} > 0$ and $n_{k+1}^{t+1} = 0$. Therefore, using Lemma F.2, treated individuals at $t+1$ should be among $\mathcal{A}_k^{t+1}$ or no one will be treated.

We next prove the second part of the lemma. If $\mathcal{I}^t \subset \mathcal{A}_k^t$, we have $\bar{n}_k^t > 0$ and $n_{k+1}^t = \bar{n}_{k+1}^t = 0$. Then Lemma F.1 implies $n_{k+1}^{t+1} > 0$ and $n_{k+2}^{t+1} = 0$. Therefore, using Lemma F.2, treated individuals at $t+1$ should be among $\mathcal{A}_{k+1}^{t+1}$ or no one will be treated. $\qquad\square$

