# OpenReview forum: "The Hidden Cost of Waiting for Accurate Predictions"
_ICLR.cc/2025/Conference — ICLR 2025 Oral_

### Official Review · Reviewer_S7QU · 2024-10-29

**Soundness:** 2
**Presentation:** 2
**Contribution:** 2
**Rating:** 8
**Confidence:** 2

**Summary:**

The paper studies the problem of resource allocation by considering the trade-off between allocation efficiency and gathering further observations. In particular, the problem is relevant in the context of societal resource allocations, where we are tasked to identify the individuals to intervene (e.g., housing benefits). The authors propose a theoretical framework to study this issue and show that waiting to acquire more observations indeed increases the predictive accuracy, but it also reduces the average ranking loss. Lastly, the authors validate their theoretical findings on a simple semi-synthetic experiment.

**Strengths:**

The authors study an interesting problem related to allocating resources to maximize the overall utility, by trading-off acquiring information to improve the predictive accuracy and the allocation timing. This topic is not in my main research area, so I cannot comment on the novelty of the approach.

**Weaknesses:**

The authors tackle this problem from a statistical point of view, rather than the classical ML formalization. Thus, the theoretical framework might be considered too simple and lacking grounding in a realistic scenario. The authors simplify and put many assumptions in place to derive their theorems and bounds, but I believe some of them might not be considered reasonable. Some examples:
- $o^t_i$ is essentially a binary variable. A more realistic assumption would have been to consider $o^t_i$  is a feature vector $x \sim P(X)$.
- $\tilde{p}$ is an increasing function. Do you have any examples about situations where it should be the case?
- $\tilde{p}$ are assumed to be known, while in practice we might have access only to an (reasonably good) estimator.

The experiment's description could be expanded. It is not clear what is meant by “failure”, or “inequality” in the context of students. Moreover, the observation model is also not clear. Providing more examples with other realistic datasets might strengthen the overall message.

Algorithm 1 is not “efficient” in a computational sense. As shown by Lemma 5.2 and Theorem 5.3, the complexity is exponent in time $T$. Even if $T$ is manageable in a real setting (references?), I would not consider it as “efficient” in the broad sense (e.g., polynomial in $T$) and I would suggest the authors rephrase it in the manuscript.

(Very) Minor nitpicking:
- I would have preferred to see a small “related work” section in the main paper, citing at least the most important/relevant works in the area. It helps position the paper in the literature, and it helps unfamiliar readers to understand the relevance of the contribution.

**Questions:**

I have no questions for the authors.

---

> ### Author Response · Authors · 2024-11-20
> **Part 1/2**
>
> We thank the reviewer for their insightful comments and for offering a fresh perspective on this problem. We agree that using statistical and machine learning tools to allocate resources presents a rich intersection of techniques from multiple disciplines. We appreciate the reviewer’s viewpoint and the different lens through which to study this issue. In the following, we discuss how our view is connected to the classic ML view and then discuss our modeling assumptions.
>
> A risk predictor involves two types of uncertainty. First, the risk predictor may not be the optimal predictor for failure, given observations from the individual. This epistemic uncertainty, also known as the generalization gap, decreases as the planner collects more historical data and fits a better model. This is the focus of classical learning theory. Second, even with the Bayes-optimal predictor, the planner cannot fully resolve the uncertainty about an individual due to the limited number of observations available for that individual. This uncertainty can only be reduced by waiting to gather more information. The focus of our study is on the dynamics of this type of uncertainty and the tradeoff it introduces. But we agree that the first uncertainty is also an interesting and fundamental question.
>
> With this context, we next discuss our modeling assumptions.
>
> > Re. $\tilde{p}$ is an increasing function. Do you have any examples about situations where it should be the case?
>
> We interpret each observation as a signal about the unknown failure probability of an individual. Examples include a student passing or failing an exam, or a patient receiving a positive or negative lab result. It is natural to assume that a positive observation is positively correlated with the failure probability; equivalently, $\tilde{p}$ is an increasing function. In the examples above, this means that a student at greater risk of dropping out is more likely to fail an exam, or a patient at higher risk of diabetes is more likely to receive a positive test result.
>
> > Re. $\tilde{p}$ are assumed to be known, while in practice we might have access only to an (reasonably good) estimator.
>
> We would like to clarify that, as discussed in line 162 and more formally in Proposition E.4, under our binary observation model and assuming a monotone $\tilde{p}(\cdot)$, the optimal ranking simply involves sorting individuals based on their number of positive observations. So we do not assume that the planner knows $\tilde{p}$. However, in analyzing the ranking and allocation, we do study how the functional form of $\tilde{p}$​ influences the conclusions.
>
> > Re. $o_i^t$ is essentially a binary variable. A more realistic assumption would have been to consider $o_i^t$ is a feature vector $x \sim P(x)$.
>
> We agree with the reviewer that supposing binary observations is an assumption that was necessary to make our study of the dynamics of optimal ranking tractable. Though it is a stylized assumption, we believe it remains applicable in a number of real-world settings or serves as a reasonable approximation. For instance, medical lab results are often interpreted as binary outcomes, and a lot of key information about students, e.g., attendance or disciplinary action, is collected as binary information. Further, information like test scores is often reduced to pass/fail or other coarse information. That being said, we agree with the reviewer that enriching how the planner collects information is interesting, and we hope that with the foundations that this paper provides, it will be an area of further exploration.
>
> We discuss the other concerns raised by the reviewer in the followup comment.

---

> ### Author Response · Authors · 2024-11-20
> **Part 2/2**
>
> > Re. The experiment's description could be expanded. … more examples with other realistic datasets might strengthen the overall message.
>
> We thank the reviewer for this suggestion. **We have updated the paper to clarify** that, in this experiment, failure refers to dropping out of school, which is recorded in the data. We also define inequality as the smallest $G$ such that the distribution of dropout probabilities is $G$-decaying. While we agree that additional experiments could further enrich the paper, our goal in Section 5 is to demonstrate that our algorithm can find the optimal allocation in a realistic setting. Even in the simplest non-contrived settings, the tradeoff between gaining more observations and losing vulnerable individuals is evident. We thank the reviewer for this suggestion and believe that further empirical work is an excellent direction for future research.
>
> > Re. Algorithm 1 is not ``efficient'' in a computational sense … I would suggest the authors rephrase it in the manuscript.
>
> We appreciate the reviewer for pointing this out. Our focus was on efficiency in terms of its independence from the number of individuals, which is often a significant bottleneck in resource allocation problems, as well as from the budget. However, we agree with the reviewer that the dependence on $T$ is also important. As a result, **we have revised the language around efficiency in the updated version, using more precise terminology (e.g., lines 343 and 380)**.
>
> > Re. Minor nitpicking
>
> We completely agree with the reviewer. In the current manuscript, we prioritized providing a comprehensive review of related work in the appendix rather than a brief one in the main text. We will certainly consider including a more concise version of the related work in the final version and appreciate the reviewer for highlighting this.

---

> ### Comment · Reviewer_S7QU · 2024-11-22
>
> I thank the authors for their answers to my review. All my concerns and questions have been addressed and resolved. After reading the other reviews and the corresponding rebuttals, I have updated my score accordingly.

---

### Official Review · Reviewer_wWAe · 2024-11-02

**Soundness:** 4
**Presentation:** 4
**Contribution:** 3
**Rating:** 8
**Confidence:** 5

**Summary:**

The authors study the tension between allocating resources based on early, noisy observations versus waiting for additional observations, with the risk of individuals dropping out or "failing." The paper focuses on (i) how ranking loss changes with more observations and (ii) identifying the optimal timing for resource allocation. Section 3 analyses the variation in ranking loss with increasing observations, Section 4 finds the optimal timing for a full-resource allocation, and Section 5 extends this to multi-step, budgeted allocations, introducing an algorithm to compute the optimal over-time allocation.

**Strengths:**

- Sections 4 and 5 are well organised, covering one-time allocation and then extending this to over-time allocation.
- The experiments in Section 5.3, with visualisations for one-time and over-time allocations offers useful insights on how larger budgets allow earlier allocations.

**Weaknesses:**

- Algorithm 1 scales exponentially with $T$; while this improves over naïve iteration, calling it “efficient” might be misleading. The experiments with NELS data used small $T$ values, but $T$ could be large with fine-grained data. Can you discuss practical limitations with larger T values?
- (Minor, organisational) Section 3 could benefit from a restructuring:
   1. Thoerem 3.1 can be moved to just before "approximating ranking risk", with a sentence on how the ranking risk can improve only if the change in population from failure is less than the gain in observations.
  2. The computations for “dynamics of ranking risk” were tedious to parse and could be moved to the appendix in the interest of readability.
- (Minor, related works) The paper does a good job covering related works in the appendix. The authors reference Abebe et al. (2020), which investigates optimal subsidy allocation to minimize failure probability. A useful addition to this line of work is Heidari & Kleinberg [1] and Acharya et al. [2], which examine welfare-optimizing policies under finite time horizons to support low-income groups. Additionally, Azizi et al. [3] (2021) on safe exits for homeless youth could complement Kube et al. (2023), which is already cited.

[1] Allocating Opportunities in a Dynamic Model of Intergenerational Mobility

[2] Wealth dynamics over generations: Analysis and interventions

[3] Designing fair, efficient, and interpretable policies for prioritizing homeless youth for housing resources

**Questions:**

Please address the first bullet point in the weaknesses section.
Also, the runtime for Algorithm 1 is with the general utility class (Sec 4.2); could it be faster with fully effective treatments?

---

> ### Author Response · Authors · 2024-11-20
>
> We thank the reviewer for their insightful review and helpful feedback. In the following, we address the points and questions raised by the reviewer.
>
> > Re. [Calling Algorithm 1] ``efficient'' might be misleading. … Can you discuss practical limitations with larger $T$ values?
>
> We appreciate the reviewer for pointing this out. Our focus was on efficiency in terms of its independence from the number of individuals, which is often a significant bottleneck in resource allocation problems, as well as from the budget. However, we agree with the reviewer that the dependence on $T$ is important. **We have revised the language around efficiency in the updated version, using more precise terminology (e.g., lines 343 and 380)**. While we typically consider time scales of months or years in social contexts, we acknowledge that at finer time scales, our algorithm may not be efficient, and a standard policy learning approach, such as reinforcement learning, would be more appropriate. We thank the reviewer for highlighting this point and helping to refine our language.
>
> > Re. (Minor, organisational) Section 3 could benefit from a restructuring
>
> This is a great suggestion! We agree with the reviewer that presenting the main result earlier in this section enhances clarity. Accordingly, **we have restructured Section 3** to present the main result immediately after introducing ranking risk, followed by a discussion of the steps leading to its proof. We appreciate the reviewer’s input and believe this change has improved the readability of our paper.
>
> > Re. (Minor, related works)
>
> We thank the reviewer for pointing out additional related works. Azizi et al. is indeed a valuable complement to Kube et al. We also agree that our proposed dynamic is closely related to dynamic models of opportunity allocation (Heidari et al.) and the dynamics of wealth across generations (Acharya et al.). **We have updated our literature review to include the suggested papers and appreciate the reviewer’s helpful input!**
>
> > Re. The runtime for Algorithm 1 is with the general utility class, could it be faster with fully effective treatments?
>
> This is an excellent question that we spent considerable time contemplating. Unfortunately, while we could improve the dependence on $T$ to something like $T/2$, as suggested by Theorem 4.3, we cannot eliminate the exponential dependency on $T$ when solving for the exact solution. Therefore, we maintain the presentation of Algorithm 1 in its most general way. We thank the reviewer for this insightful question, which prompted us to rethink this aspect!

---

### Official Review · Reviewer_qU3b · 2024-11-03

**Soundness:** 4
**Presentation:** 4
**Contribution:** 4
**Rating:** 8
**Confidence:** 4

**Summary:**

The paper presents a stylistic framework for analyzing the utility of intervention strategies when information about individuals is revealed in an online / stepwise manner. In particular, suppose a mechanism designer has access to some intervention budget $B$ over a set of _active_ individuals $\mathcal{A}$. At each time step $t$, each individual $i$ may drop out of the active pool $\mathcal{A}_t$ based on sampling from a Bernoulli distribution with probability $\tilde{p}_i$. The goal of the mechanism designer is to intervene with the high risk individuals (those at high risk of dropping out of the active pool). The challenge is that there is a tradeoff between the available amount of information on each individual (which improves via waiting for more time to pass), and the effectiveness and/or ability to intervene. The latter is impacted by two facts (1) high risk individuals benefitting most from interventions may drop out of the pool earlier; and (2) utility / welfare is higher when intervening at individuals earlier in the time horizon (the authors assume a concave welfare utility modeling function based on individual probabilities $p_i$).

With the general framework setup, the authors tackle three specific settings.
1. Section 3: Bounding the _ranking risk_ for ranking all individuals at each time step based on their probability of dropping out of the active pool.
2. Section 4: When the mechanism designer can intervene upon the $B$ individuals most at risk, but only at one specific point in time, how do we find the optimal point, and how good is it?
3. Section 5: If the mechanism designer can spread out its intervention budget over the time horizon (i.e., intervene at different points in time), how can the optimal strategy be calculated / computed in a tractable manner?

To adress the first point, the authors derive an exact characterization of when the ranking risk improves with information collection (Theorem 3.1). At a high level, this charectizeration shows that the ranking risk improves only if the impact of individuals dropping out of the population is dominated by some function of the information gain, modulo constant factors.

To address (2), the authors show in Theorem 4.3 that (under the fully effective treatment assumption) the best time to intervene with the entire budget $B$ depends on the total time horizon $T$, the number of individuals, budget, and amount of inequality amongst the individuals (captured by a parameter $G$ for a $G$-decaying distribution). Higher inequality means that the intervention must be applied earlier in order to be most effective. A similar characterization is made for the more general setting where interventions are not 100% effective, but decay with time or depend on the underlying probabilities (Theorem 4.5).

Finally, the authors move to the more complex setting where the budget may be distributed across different points of time $t$. They demonstrate that a naive approach of computing the best policy in the resulting MDP is intractable, but by using particular structure of the problem, can be simplified into searching over a much smaller number of parameters (Theorem 5.1). That is, the optimal intervention will have a specific structural form, whose parameters can be efficiently optimized over (Lemma 5.2).

Lastly, the authors run some experiments on real data from the national eduational longitudinal study. An important takeaway echoed in each of the sections is that it is often beneficial to intervene earlier — with noisier data — than it is to wait until better information is collected.

**Strengths:**

This paper is extremely well-written, concise and clear, and also makes progress on a very important problem: how does information gain (in terms of improving accuracy of machine learned predictors) trade off with the potential harms / opportunity costs of intervention delays?

Originality: The stylistic framework seems reminiscent of an over time version of Shirali et al. — who ask about the utility of individual predictions when faced with intervention budgets — but most technical details are different. I believe the online ranking / budgeting setting of this work is quite natural, realistic, and novel.

Clarity: I found the writing precise and clear throughout.

Significance: Although the results are within a stylistic framework, the paper presents another datapoint (in addition to Shirali et al.) supporting the fact that accurate individual level predictions may not or should not be the end-all be-all goal when maximizing the effectiveness of interventions in school, healthcare, etc. These works together are surprisingly counter-intuitive, given that most of the work within the areas of algorithmic fairness and individual level predictions focuses on obtaining accurate predictions. This paper suggests that when viewed from a higher level within the _context_ of decision making and budgeted intervention, the accuracy of individual level predictions may matter far less than one might think. Overall, this work has the potential to guide much future research in the direction of what is actually most impactful within the context of real decision making and intervention systems. Because of this, I believe this work will be highly significant within the next few years.

I also agree with the authors and think there are numerous potential directions that can build upon this work. For example, the fair ranking community has proposed alternatives to simply ranking individuals by their probability of dropping out (e.g., Singh et al. 2021). Whether or not this is the correct thing to do is certainly context dependent, but one can imagine similar characterizations for when information gain may hurt or help when applying interventions in non-welfare maximizing ways (in the interest of fairness).

Singh et al. 2021: Fairness in ranking under uncertainty.

**Weaknesses:**

The only weaknesses I have are minor, although I have not carefully checked the proofs of the statements. First, I believe that the independence in Assumption 4.2 and line 102 should be discussed more. In particular, imagine the mechanism designer is potentially intervening on a pool of students who are all enrolled in a particular “catch-up” class for low performing students. Since all students have the same teacher, we may expect that their observations may be correlated. For example, if the teacher is really good, then maybe nobody drops out of the pool and no intervention is necessary (the opposite also holds true).

Similarly, I think assumption 4.4 can also be discussed in more detail. I understand that this is a technical assumption, but perhaps a note in the appendix about what kind of utilities this can capture, or some common examples, may be useful. I may have missed this somewhere though!

Minor comments
1. Typo: line 807 “particular, Our model”
2. Are there possible citations for line 266-267: “For instance, consider housing vouchers or dropout prevention programs, which have been found to be very effective.”?

**Questions:**

How should I think about $\gamma$ in Assumption 4.2 and the reliance on $\gamma$ for Theorem 4.3?

---

> ### Author Response · Authors · 2024-11-19
>
> We are heartened to read the reviewer’s summary of our paper and their perspective about its potential feedback!
>
> > Re. ... there are numerous potential directions that can build upon this work. For example, … one can imagine similar characterizations for when information gain may hurt when applying interventions in non-welfare maximizing ways.
>
> We thank the reviewer for highlighting this excellent potential avenue for future research. We agree that the tradeoffs between acting early and waiting to reduce uncertainty extend beyond our utilitarian framework. Singh et al. provide the right framework for considering uncertainty in fairness-sensitive settings, and we believe our dynamic model can be extended in this direction. **We appreciate this suggestion and have included it into our updated discussion**.
>
> > Re. … the independence in Assumption 4.2 and line 102 should be discussed more. In particular, imagine the mechanism designer is potentially intervening on a pool of students.
>
> We thank the reviewer for this insightful suggestion. In our framework, we avoid imposing additional structure to the problem by assuming independence of observations. However, we fully acknowledge that these assumptions may not hold in problems with additional structure, such as the example provided by the reviewer. Specifically, we rely on two key assumptions: (1) failure events are independent, which does not apply in scenarios where students share the same teacher, for instance; and (2) intervention effects are independent, meaning there are no spillover effects. **We appreciate the reviewer highlighting these points and have clarified the implications of these assumptions in the updated manuscript**. We will discuss Assumption 4.2 in the following in response to another question from the reviewer.
>
> > Re. … assumption 4.4 can also be discussed in more detail. … perhaps a note in the appendix about what kind of utilities this can capture, or some common examples, may be useful.
>
> We thank the reviewer for this excellent suggestion! We agree that including additional examples illustrating $(\lambda_1, \lambda_2)$-decaying utilities would improve the paper's clarity. **Therefore, we added a few more examples, such as when the treatment is partially effective, under a new Proposition E.11**, and have referenced this proposition in the main text. We believe these additions improve the clarity of our work and appreciate the reviewer’s valuable input!
>
> > Re. How should I think about $\gamma$?
>
> We thank the reviewer for the clarifying questions. The introduction of $\gamma$ in the observation model was primarily to simplify certain proofs without explicitly imposing Lipschitz continuity or bounding the curvature of $\tilde{p}(\cdot)$. So, it is more of a proof artifact than a fundamental aspect of the model. That being said, in most proofs, a large gamma implies that individuals with small $p$ may have highly distinct observation probabilities, making them easier to distinguish, while individuals with large $p$ tend to have more similar observation probabilities, making them harder to differentiate. This often encourages making additional observations to better identify individuals in need. However, we emphasize that this interpretation is more of an intuitive explanation rather than a formal claim, as $\gamma$ is mainly a technical construct.
>
>
> > Re. minor comments
>
> We fixed the typo and cited the significant positive effect observed from allocating housing vouchers to homeless families in line 269. We thank the reviewer for these suggestions!

---

> > ### Comment · Reviewer_qU3b · 2024-11-20
> >
> > I thank the reviewers for the revisions in the draft and for addressing my comments. I believe the work is of high quality and will certainly be built upon by others. I don't see the simplicity of the model / stylistic setting to be a drawback, if anything (in my opinion) this is a feature since it makes the work very accessible. The work develops certain intuitions about trading off information gain with quick decision making, which I believe are broadly applicable.

---

### Official Review · Reviewer_XpSw · 2024-11-04

**Soundness:** 4
**Presentation:** 3
**Contribution:** 4
**Rating:** 8
**Confidence:** 3

**Summary:**

This paper considered resource allocation based on predictions of ranking among individuals. The resources are interventions to prevent individuals from experiencing undesirable outcomes. The authors examined the tension between relying on earlier, possibly noisier predictions to intervene before undesirable outcomes, and waiting to intervene with more precise allocations after gathering more observations.  Through statistical analysis, they showed that individual prediction accuracy could improve with more observations, but the overall ranking performance does not necessarily improve. Moreover, they identified inequality, which is the variance of individual failure probabilities, as a driving factor leading to this counterintuitive behavior. When the planner needs to allocate resource at once, an upper bound on the optimal allocation time was given in the paper. When the planner can allocate resources over time, an algorithm that is provably optimal with respect to the total utility is developed.

**Strengths:**

1. The paper studied a novel problem of ranking prediction guided resource allocation. The connection between prediction quality and downstream decision performance is an important concept that is not well understood yet. The paper contributed new insights towards the gap.

2. The paper highlighted interesting tradeoffs between waiting for observations to improve prediction accuracy and losing vulnerable individuals due to waiting. The connection to inherent inequality in the population further enriched these results.

3. The paper provided rigorous theoretical derivation of all the results. The problem formulation, despite necessary simplification, is general enough to represent broad application contexts.

**Weaknesses:**

1. The visualizations in Section 5.3 effectively illustrate the theories discussed in the section on sequential allocation. Sections 3 and 4 would benefit from similar empirical support. Incorporating experiments or even basic numerical examples would make the theoretical results more accessible and intuitive. For example, in Section 4, an experimental validation of the derived upper bound on the optimal timing would offer a concrete sense of how these bounds apply in practice.

2. The writing in Sections 3, 4, and 5, while necessarily technical, sometimes read dense. It would be helpful to have clearer explanations of the formulas. In addition, discussing the intuition and the logic of complex derivations behind key theorems and algorithm would also help with the overall flow and clarity.

3. Additional explanations could clarify specific aspects of the formulation. For instance, the paper focused on predicting failure probabilities as a basis for resource allocation; however, it might also consider scenarios in which qualification probabilities are predicted instead, and resources are allocated based on those rankings. A discussion on why the study emphasized failure probability as opposed to other potential metrics would be informative. In addition, resource allocation problems often involve various constraints beyond following the ranks among individuals. A discussion of whether such constraints could be incorporated—and if not, why they were excluded—would add to the completeness of the paper.

**Questions:**

1. In the derivation of Section 4.1, a measure of inequality is introduced in Definition 4.1. What are the connections, if any, between this measure of inequality and the variance of individual failure probabilities as adopted in Section 3?

2. In the studied budget allocation setup, are all individuals assumed to require the same budget? If so, can these budgets be viewed as available slots to an opportunity, e.g. the number of students that can be enrolled in a support program, and will assigning different budgets to different individuals affect the presented results?

---

> ### Author Response · Authors · 2024-11-19
> **Part 1/2**
>
> We thank the reviewer for their detailed feedback and insightful questions. We are pleased to read the reviewer’s appreciation about the novelty of the questions around prediction and downstream decision performance, as well as the broad applicability of the insights.
>
> Below, we first address the reviewer’s questions and then outline other improvements made to the paper in response to the feedback.
>
> > Re. Q1: What are the connections, if any, between the measure of inequality [in Section 4] and the variance of individual failure probabilities as adopted in Section 3?
>
> We are grateful to the reviewer for this question, which highlights a tight relationship that exists between our formulation of inequality and the variance. We were intrigued by this question and spent quite a bit of time thinking about this connection. **In the updated version of the paper, we provide a precise characterization of the connection between our notions of inequality in the new Proposition E.10**. Specifically, we present a tight lower bound for the variance in terms of $G$ and demonstrate that this lower bound decreases with $G$. In other words, a low value of $G$, which corresponds to high inequality, guarantees a high variance. Therefore, we could also present a slightly weaker version of Theorem 3.1 in terms of $G$ instead of $\text{Var}^t[p]$. We thank the reviewer again and believe these new observations have enriched our paper!
>
> > Re. Q2: … are all individuals assumed to require the same budget? … will assigning different budgets to different individuals affect the presented results?
>
> Yes, this is correct. Intervening on any individual incurs a unit cost. In our results, we aimed to avoid imposing additional structure on the intervention costs to highlight that these counterintuitive insights arise even in such simple settings. Also, some generalizations of the cost structure may already be implicitly captured in a version of our problem. For example, if it costs $c(p)$ to intervene on an individual with a failure probability of p, as long as $u^t(p)/c(p)$ remains monotone increasing in $p$, we can still use predictions of $p$ to optimally rank and allocate. However, when monotonicity is broken, we can no longer make simple arguments about the optimal ranking. We believe these are promising directions for follow-up work from the research community.

---

> ### Author Response · Authors · 2024-11-19
> **Part 2/2**
>
> > Re. W1: The visualizations in Section 5.3 effectively illustrate the theories discussed. … Incorporating experiments or even basic numerical examples would make the theoretical results [in Sections 3 and 4] more accessible and intuitive.
>
> We are glad that the reviewer found the visualization in Section 5 helpful. We agree that a similar illustration can benefit other sections, particularly Section 4. To further clarify the role of budget size and inequality in deriving the theoretical results of Section 4, **we have added a one-page illustrative example in Appendix C, including an extensive discussion and visualization**. This example intuitively demonstrates why high inequality, reflected in a low $G$, or a large budget, may favor earlier one-time allocations. We thank the reviewer for this valuable suggestion and believe the new illustration has enriched our paper!
>
> > Re. W2: The writing in Sections 3, 4, and 5, while necessarily technical, sometimes read dense.
>
> We appreciate the reviewer’s suggestion to provide further intuition and summary to enhance the clarity of the results. **We have improved the readability and clarity of the paper**, in particular, we have restructured Section 3 to present the main result earlier, provided an illustration for Section 4, and further elaborated on Definition 4.2. If there are any remaining parts the reviewer recommends revisiting, we welcome any further thoughts.
>
> > Re. W3: Additional explanations could clarify specific aspects of the formulation. … [The paper] might also consider scenarios in which qualification probabilities are predicted instead.
>
> To make sure we understand the reviewer’s question: We assume that by a qualification probability, the reviewer means a metric like student GPA, job applicant score, health measure, and so on rather than predicting school dropout, job retention, or hospital admission.
>
> In this case, if the planner’s objective is still to prevent poor outcomes, then they may use these qualification metrics: e.g., by setting some threshold on these metrics, to predict poor outcomes. In this way, these metrics serve as a proxy. In our work, we formulated the predictions to use all available information at a time to make the best possible prediction, so limiting to a single proxy like GPA would only weaken the planner’s prediction and our insights would hold even more strongly.
>
> If the planner’s objective is different than preventing poor outcomes, e.g., if they instead want to maximize the average GPA, then we agree with the reviewer that our framework does not easily map to this setting and indeed this is an entirely different set of questions that would require defining different objectives and problem formulations around, e.g., the effect of allocations.
>
> In our work we focus on the prevention of poor outcome problems as that also is well-studied and well-motivated. We agree however that there may be similar insights that might hold in this continuing setting with different objectives and believe this too could be a promising area for exploration.
>
>
> > Re. W3 (Cont.): … resource allocation problems often involve various constraints beyond following the ranks among individuals. A discussion … would add to the completeness of the paper.
>
> We also agree with the reviewer that there are ways to enrich the model further, such as by considering various constraints beyond just the ranks of the individuals. One example would be to introduce welfare weights among the individuals, allowing the policymaker to prioritize one individual over another, even if they have the same failure probabilities. We could also consider other constraints: for instance, in the context of over-time allocation, there may be barriers to concentrating expenditures around the same time point. It is straightforward to adapt our results to some of these constraints, e.g., adding welfare weights, whereas others ought to be areas for future exploration. As the reviewers point out, since our insights hold even in this simplified version of the model, additional constraints may make the tradeoffs we highlight here even more pronounced, though this requires deeper investigation. **We have updated our discussion section to provide a more extensive discussion around these and other promising directions highlighted by the reviewer**.

---

> > ### Comment · Reviewer_XpSw · 2024-11-24
> >
> > I would like to thank the authors for their detailed response to my comments and questions. I believe the changes mentioned in the authors' responses improve the paper. Therefore, I have updated my scores accordingly.

---

### Author Response · Authors · 2024-11-20

We thank the reviewers for their deeply engaged, thorough, and constructive reviews. We found the detailed reviews and insightful questions highly encouraging. The feedback has greatly enriched our work.

Below, we address each of the questions raised by the reviewers and have uploaded a new version of the work, in line with the reviewers’ feedback. We have highlighted major changes to the text in blue.

---

### Meta-Review · Area_Chair_grGP · 2024-12-16

**Metareview:**

This paper studies a resource allocation problem in a pool of individuals where waiting for more observations improves resource allocation. The catch is that some individuals may leave the pool if their resources are not allocated on time. The authors propose a mathematical model of the problem and study it in two resource allocation settings (one time and online). The proposed algorithms are also evaluated empirically. The paper is well written and executed. Its scores are 4x 8, which is a major improvement over the initial 8, 2x 6, and 5. This is a clear accept. Both the reviewers and I believe that this paper touches on several timely topics (fairness and delayed feedback), and therefore should be highlighted at the conference.

**Additional Comments On Reviewer Discussion:**

See the meta-review for details.

---

### Decision · Program_Chairs · 2025-01-22

Accept (Oral)